# Selection of start codon during mRNA scanning in eukaryotic translation initiation

Ipsita Basu[1], Biswajit Gorai [1,3,5], Thyageshwar Chandran [2,4,5], Prabal K. Maiti [1✉] & Tanweer Hussain [2✉]

Accurate and high-speed scanning and subsequent selection of the correct start codon are important events in protein synthesis. Eukaryotic mRNAs have long 5′ UTRs that are inspected for the presence of a start codon by the ribosomal 48S pre-initiation complex (PIC). However, the conformational state of the 48S PIC required for inspecting every codon is not clearly understood. Here, atomistic molecular dynamics (MD) simulations and energy calculations suggest that the scanning conformation of 48S PIC may reject all but 4 (GUG, CUG, UUG and ACG) of the 63 non-AUG codons, and initiation factor eIF1 is crucial for this discrimination. We provide insights into the possible role of initiation factors eIF1, eIF1A, eIF2α and eIF2β in scanning. Overall, the study highlights how the scanning conformation of ribosomal 48S PIC acts as a coarse selectivity checkpoint for start codon selection and scans long 5′ UTRs in eukaryotic mRNAs with accuracy and high speed.

[1] Center for Condensed Matter Theory, Department of Physics, Indian Institute of Science, Bangalore 560012, India. [2] Department of Molecular Reproduction, Development and Genetics, Division of Biological Sciences, Indian Institute of Science, Bangalore 560012, India. [3]Present address: Department of Chemical Engineering, University of New Hampshire, Durham NH-03824, USA. [4]Present address: Department of Biotechnology, National Institute of Technology-Warangal, Telangana 506004, India. [5]These authors contributed equally: Biswajit Gorai, Thyageshwar Chandran. ✉email: maiti@iisc.ac.in; hussain@iisc.ac.in

Eukaryotic translational initiation is a complicated yet well-coordinated process, where the pre-initiation complex (PIC; the complex of 40S ribosomal subunit and initiation factors) binds to the 5′ end of mRNA and scans along until a start codon is encountered in the ribosomal P site[1]. In brief, the overall event starts with the initiation factors eIF1, eIF1A, and eIF3 binding to the 40S subunit, which facilitates the binding of charged initiator tRNA (Met-tRNA$_i$) as a ternary complex (TC) with eIF2-GTP[1,2]. The factor eIF5, a GTPase activating protein (GAP), is recruited along with TC or eIF3[1,3–5]. The 43S PIC thus formed is recruited to the capped 5′ end of mRNA with the help of eIF4 factors[6,7] leading to the formation of the 48S PIC. This 48S PIC then scans the 5′ untranslated region (UTR) of the mRNA for the cognate start codon in an open (P$_{OUT}$) conformation. The initiation factors eIF1 and eIF1A are known to stabilize the PIC[8,9]. The TC in the P$_{OUT}$ state allows tRNA$_i$ to inspect successive triplets of mRNA nucleotides entering the P site for complementarity to the anticodon. The GTP bound to eIF2 may be hydrolyzed during the scanning process; however, the phosphate (P$_i$) is not released[1,7,10]. Upon recognition of the start codon, the PIC undergoes conformational changes to form a scanning-arrested closed (P$_{IN}$) complex accompanied by the release of eIF1, which is essential for the fidelity of start codon selection, and dissociation of P$_i$[2].

In contrast to eukaryotes, prokaryotes use a simpler mechanism that permits initiation from UUG and GUG apart from AUG. Also, there is no scanning mechanism in prokaryotes, which use the nearest available AUG to 5′ end of mRNA as the start site with the upstream Shine Dalgarno (SD) sequence. Studies have indicated that a kink in mRNA between the A and P site plays an important role in maintaining the stability of codon-anticodon interaction, allowing the selection of AUG as the start codon. On the contrary, the eukaryotic system involves several initiation factors. Also, eukaryotic genes are characterized by long 5′ UTRs that can exceed 1000 nucleotides; for instance, in humans, the maximum length reported is 2803[10–12]. The median length of 5′ UTR is ~53 nucleotides in budding yeast and 53–218 nucleotides for higher organisms[12]. A recent finding shows that only one ribosome scans a 5′ UTR at a time in most human cells, and the length of 5′ UTR affects translation efficiency[13]. Hence, it is essential to address how the ribosomal PIC is able to scan long 5′ UTR with high speed and accuracy.

To recognize the start codon, the 48S PIC has to accurately inspect successive triplets of mRNA nucleotides entering the P site at high speed. The process is unidirectional and in base-per-base mode[14]. The net speed of scanning was found to be about 8–10 nucleotides per second in cell-free extract and the scanning rate is expected to be even higher within the cell[14,15]. However, there is no clear understanding of the mechanism of scanning and how the 48S PIC can scan at such a high speed with accuracy.

It is assumed that the 48S PIC would continue to scan the 5′ UTR in an open conformation until it reaches the start codon. Recognition of the start codon leads to scanning arrest and conformational rearrangement to the closed state of the 48S PIC[5,16]. Further, the downstream events in translation initiation are triggered following the release of eIF1[1,5,17–20]. Alternatively, it is suggested that the 48S PIC may shuttle spontaneously between open (P$_{OUT}$) and closed (P$_{IN}$) conformations during scanning[21–23]. Based on this model, the energetics of initiator tRNA (tRNA$_i$) binding to different near-cognate codons in the yeast 48S PIC in closed conformation was studied using atomistic molecular dynamics (MD) simulations and energy calculations in the presence or absence of eIF1 and eIF1A to understand scanning[21,22] using a structure of partial yeast 48S complex in P$_{IN}$ state (py48S)[20]. The results indicated that eIF1 was primarily involved in discrimination against mismatches in the first and second positions of the codon, whereas eIF1A is involved in discrimination against near-cognate codons with third position mismatches in the P$_{IN}$ state[21]. However, these results consider the energetics of codon discrimination only in a scanning-arrested P$_{IN}$ state, which cannot be extrapolated to the open scanning conformation of the 48S PIC because of the conformational differences in the two states (discussed below).

Furthermore, the simulation that was initialized from the closed-state of 48S PIC without the tRNA$_i$ was considered a model for the P$_{OUT}$ state[21]. It was assumed that there is no interaction between codon and anticodon in the P$_{OUT}$ state. However, with the availability of a cryo-electron microscopy (cryo-EM) structure of a 48S PIC in its open conformation (py48S-open complex)[24,25], it became clear that the open conformation is significantly different from the closed one (py48S-closed complex) and, importantly, codon-anticodon interaction does indeed occur in the open state.

The reported structures[24,25] revealed that, in the py48S-open complex, the 40S head moves upwards with respect to the body with attendant relaxation of rRNA helix 28, which connects the head to the body of the 40S. As a result, both the mRNA channel and the P site are widened and the mRNA latch is opened. Moreover, the tRNA$_i$ is positioned in the P site ~7 Å away from the 40S body compared to that found in the py48S-closed complex. eIF1 is observed at its primary binding position at the P site in the open state and it undergoes subtle repositioning on the transition to the closed state to accommodate tRNA$_i$ in the P$_{IN}$ conformation[24]. Strikingly, the N-terminal tail of eIF1A, which was observed in proximity to the codon:anticodon duplex in py48S-closed complex, could not be observed in py48S-open complex, highlighting its role in stabilizing specifically the closed conformation of the 48S. In addition, eIF2β contacts eIF1 in the py48S-open complex but moves away from eIF1 in the py48S-closed complex, indicating its role in stabilizing specifically the open conformation. Given these striking differences in the conformation of 40S, eIFs, and tRNA$_i$ between open and closed states of the 48S PIC, the coordinates of the closed state of the 48S PIC without the tRNA$_i$ do not accurately reflect the true open conformation.

Moreover, the observation of codon:anticodon interaction in the structure of py48S-open complex with an AUC start codon in the P site[24] indicated that the tRNA$_i$ could inspect the incoming codon in the P site even in the open conformation. This led us to consider whether the 48S PIC in an open conformation can accurately recognize the start codon and discriminate against noncognate codons while scanning the 5′ UTR. If this is indeed true, then it may provide insights into codon selection during scanning and also an explanation for the high speed of scanning as the ribosomal initiation complex would not have to undergo a large conformational change (from open to closed state and back) to inspect every single incoming nucleotide triplet in the P site.

Therefore, in this study, we decided to regard the binding energy of the tRNA$_i$ with each noncognate codon relative to the cognate AUG start codon in the open conformation of the 48S PIC as a determinant of its frequency of selection as a start codon during scanning. A similar approach was used earlier to examine the selection of noncognate start codons in the closed state[21]. Here, we report that the open conformation can recognize the start codon AUG and discriminate against most noncognate codons. Recognition of AUG in the open state seems to prepare the 48S PIC to change its conformation to the closed state. Our studies also indicate that eIF1 plays a crucial role in codon selectivity in the open conformation of the 48S PIC. However, recognition of AUG as a start codon is still inaccurate owing to the failure to discriminate against codons (GUG, CUG, UUG) with a first base-pair codon:anticodon mismatch. Hence, the open conformation of the 48S PIC serves as an initial checkpoint for

the selection of the start codon at which almost all noncognate codons are rejected. A few near-cognate codons accepted in the open state can then be re-examined in a more stringent, second checkpoint, i.e., the closed conformation of the 48S PIC. Thus, our study provides insights into how the 48S can maintain accuracy at a high rate of scanning by utilizing the open state as a coarse selectivity checkpoint to reject all but a few of the possible codon:anticodon mismatches.

## Results and discussion

**Relative binding energies of non- and near-cognate codons: a possible cue for codon selection.** The scanning of the 5′ UTR by the 48S PIC primarily involves the anticodon of tRNA$_i$ interacting with mRNA at the P site, actively encountering codon triplets probably with one, two or three mismatches compared to the cognate start codon, AUG[1]. In order to understand the mechanism of codon selection in an open conformation of the 48S PIC, we carried out an array of comprehensive MD simulations of the core region of the py48S-open-eIF3 PIC (Fig. 1 and

Supplementary Fig. 1). The simulations were carried out with cognate (AUG) as well as with multiple non-AUG codons with one, two, or three mismatches (Fig. 2, Supplementary Fig. 2 and Supplementary Data 1). The relative binding energy perturbations from the respective simulations provided insight into codon selection by the PIC in its open state. Simulations were also carried out by excluding one or more initiation factors from the system (eIF1, eIF1 + eIF1A, eIF2β, or eIF2α) to evaluate their roles in the mechanism of start codon recognition (Fig. 3 and Supplementary Data 2).

**Energetics of noncognate codons (two or three mismatches with the tRNA$_i$ anticodon).** Out of twenty-seven possible triplets with mutations from AUG at all three positions (3-point mutations), relative binding energies were calculated for two such triplets selected at random, ensuring that both transition (point mutations involving the interchanges of two ring purine base, i.e., A ↔ G, or of one ring pyrimidine base, i.e., C ↔ U) and transversion (point mutations involving change from a purine to

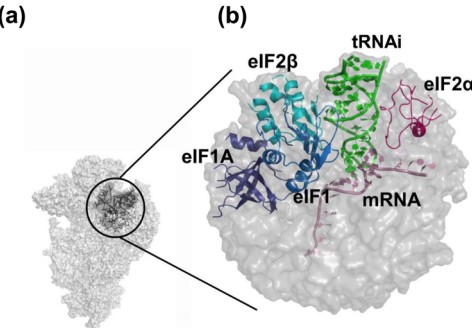

**Fig. 1 Schematic representation of the simulation sphere. a** The ribosomal 48 S PIC in open state (PDB ID: 6GSM) is shown in surface representation. The simulation sphere around P site is encircled. **b** A zoomed in view of the simulation sphere. The portions of initiation factors eIF1 (indigo), eIF1A (violet), eIF2α (purple), and eIF2β (blue), the initiator tRNA (green) and mRNA (pink) are shown in cartoon representation. The AUG codon at the P site of mRNA was used for calculating the binding energy for cognate start codon:anticodon (AUG:UAC) interactions. The codon at P site was mutated to different codons to calculate the binding energy for respective mutant codon:anticodon interactions. The relative binding energies were calculated with respect to the AUG start codon.

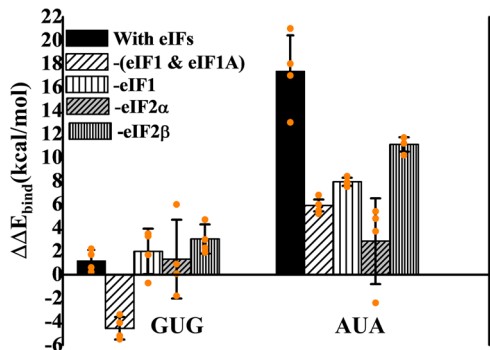

**Fig. 3 Relative binding energy profiles of near cognate (GUG and AUA) codon-anticodon interactions with respect to AUG. The simulation runs were carried out in the presence and absence of different eIFs.** For GUG and AUA, for each case, the relative binding energy is calculated with respect to AUG codon for the similar system and then the average relative binding energy is plotted as bar. The error bars indicate the standard errors obtained from the mean of relative binding energies from four independent simulation runs. AUA shows much lower relative binding energy in the absence of eIF1 alone, eIF1 and eIF1A or eIF2α. Supplementary Data 2 contains the relevant source data.

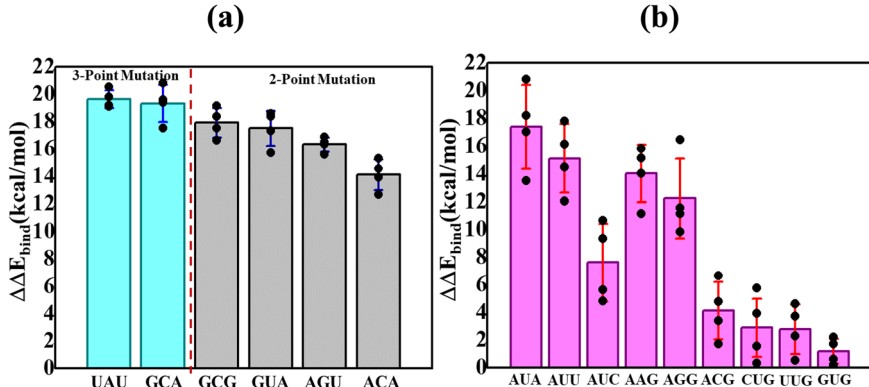

**Fig. 2 Relative binding energies of non- and near-cognate codons.** Relative binding energy profiles of (**a**) 3- and 2- point, and (**b**) 1-point mutations of the start codon. Bar chart representing the relative binding energies of 3 (turquoise), 2 (gray), and 1 (pink) point mutations calculated with respect to AUG codon. The average binding energy between codon AUG with anticodon UAC interaction is −21.4 kcal/mol, calculated using MMPBSA, over the four independent simulations. The error bars indicate the standard errors obtained from the mean of four independent simulation runs. Supplementary Data 1 contains the relevant source data.

pyrimidine base or vice versa) mutations at all three positions were represented. The calculated binding energy for cognate start codon:anticodon (AUG:UAC) interactions were used to derive the net relative binding energies, which were obtained by subtracting the respective binding energies of the noncognate states, respectively $[\triangle\triangle E_{bind} = \triangle E_{Bind}^{Mut} - \triangle E_{Bind}^{AUG}]$. The calculated relative energies for the two noncognate codons are in the range of 19 to 20 kcal/mol (Fig. 2a and Supplementary Data 1). In the average MD simulation structures with GCA and UAU, no base pairing was observed between the codon and the anticodon (Supplementary Fig 2c). We consider it very likely that other noncognate codons with 3-point mutations have similar high relative binding energies because no base pairing between codon:anticodon is expected in these noncognate codons as well. Such very high energetic penalties clearly indicate the ability of the 48S PIC to preferentially select AUG over 3-point mutations (GCA and UAU) in its open state.

We carried out similar analyses on four triplets (ACA, GUA, GCG, and AGU) with mutations from AUG at two positions, which were selected from all twenty-seven possible 2-point mutations to include transition mutations in codon positions 1 and 2 (GCG) or 1 and 3 (GUA), and both transition and transversion mutations at positions 2 and 3 (ACA and AGU). The relative energies of the 2-point mutations also have high energetic penalties in the range of 14–18 kcal/mol (Fig. 2a and Supplementary Data 1), further indicating the ability of the 48S PIC to discriminate against noncognate codons in its open state. Again, in the average MD simulation structures with 2-point mutations no base pairing was observed between the codon and the anticodon (Supplementary Fig. 2b).

The results suggest that the 48S PIC in its open conformation has the ability to reject codons with two or three mismatches with the tRNA_i anticodon, thereby rejecting a majority of the non-AUG codons encountered in the P site during scanning. This would obviate the requirement for conformational switching to the closed state in order to reject the majority of all non-AUG triplets, which might help explain how accuracy is maintained at a high speed of scanning of 5′ UTRs by the PIC.

**Energetics of near-cognate codons (one mismatch with the tRNA_i anticodon).** We next calculated the energy perturbations from simulations examining all nine possible near-cognate triplets (GUG, UUG, CUG, AGG, AAG, ACG, AUA, AUU, and AUC) with a mutation from AUG at one position (1-point mutation). Intriguingly, the calculated relative binding energies fall into two distinct groups. The first group, consisting of triplets AUA, AUU, AAG, and AGG, showed higher energy penalties in the range of 12–17 kcal/mol, while AUC showed an energy penalty of 7.6 kcal/mol (Fig. 2b and Supplementary Data 1). The remaining triplets ACG, CUG, GUG, and UUG, forming the second group, showed only moderate penalties of 1-4 kcal/mol (Fig. 2b and Supplementary Data 1). Base pairing between codon:anticodon is observed in the case of the latter group having low penalties, whereas it was absent in the former group with high energy penalties (Supplementary Fig. 2a). These results are in agreement with in vitro and in vivo studies wherein codons in the second group have been shown to support the highest levels of non-AUG initiation[26–34].

The results indicate a strong preference for 'G' in the third position of the triplet to achieve the lowest energy penalties, as any of the single-point mutations involving this base (in triplets AUA, AUU, or AUC) conferred much higher energy penalties (Fig. 2b). Moreover, the single-point transversion mutations at the second position (U) to either purine (i.e., AUG → AAG and AUG → AGG) are not tolerated and likewise confer large relative

binding energies, which was not observed for the transition mutation of U → C in the ACG triplet (Fig. 2b). Interestingly, both transversion and transition mutations at the first base seem to be tolerated (in triplets CUG, UUG, and GUG), which implies the inability of the PIC in its open conformation to efficiently recognize the first nucleotide of the AUG start codon.

In brief, the calculations suggest that all but four of the 63 non-AUG codons can be rejected by the scanning PIC in its open conformation. For the four near-cognate codons with much lower energy penalties, ACG, CUG, GUG, and UUG, the 48S PIC may then proceed to the closed conformation and execute a second accuracy check in order to reject these near-cognate triplets and achieve stringent AUG selection.

**Energetics in the presence and absence of initiation factors.** To gain insight into the roles of initiation factors in start codon recognition in the open conformation of the 48S PIC, we carried out three sets of simulations in the presence and absence of particular eIFs, including eIF1, the combination eIF1 and eIF1A, eIF2α and eIF2β, in each case using mRNA with cognate (AUG), first-position near-cognate (GUG), or third position near-cognate (AUA) start codons. It may be noted that GUG shows a low energy penalty, whereas AUA falls in the high energy penalty group (Fig. 2b).

In the absence of eIF1, the relative binding energy for AUA was found to be 7.9 kcal/mol (Fig. 3 and Supplementary Data 2), much less than that in its presence (17.4 kcal/mol). For GUG, in contrast, the relative binding energy of only ~1 kcal/mol over AUG was observed in the presence of eIF1 and a similar value of 2 kcal/mol was observed in the absence of eIF1 (Fig. 3 and Supplementary Data 2). This suggests that eIF1 has a role in the discrimination against the third position near-cognate AUA triplet in the open PIC, such that the energy penalty for AUA versus AUG is reduced in the absence of eIF1. By contrast, eIF1 does not seem to discriminate against the first-position near-cognate GUG triplet in the open PIC, which might help to explain the minimal energy penalty for GUG in the open PIC with eIF1 present.

eIF1 is known to increase the fidelity of start codon selection by opposing both transition to the closed state and subsequent P_i release at non-AUG codons[5,35,36]. This is achieved through its interaction with eIF2β exclusively in the open complex[24] and by imposing a steric block that prevents the codon:anticodon duplex from achieving the P_IN conformation[20,24]. The latter provokes displacement of eIF1 from its original position at the P site on AUG recognition in the closed complex as a prelude to its subsequent dissociation from the PIC[17,20,24,37]. As a result, eIF1 mutations that decrease its abundance, weaken its interaction with eIF2β or the 40S subunit or diminish its clash with tRNA_i, all allow inappropriate rearrangement to the closed state at near-cognate UUG codons in vivo[24,38–40]. In the open conformation of the 48S PIC, the mRNA channel is widened and eIF1 is observed in its original position at the P site, exerting no steric hindrance to the codon:anticodon duplex[24]; however, as discussed later, β-hairpin loop-1 of eIF1 in the P site interacts with the codon in average MD structure.

Simulations conducted in the absence of both eIF1 and eIF1A gave results similar to those observed in the absence of eIF1 alone for the AUA triplet, as the relative binding energy was reduced from ~17.4 to 6 kcal/mol, which is only slightly lower than that in the absence of eIF1 alone (~7.9 kcal/mol). These findings suggest that eIF1A also exerts some discrimination against AUA in the open PIC independently of eIF1. Surprisingly, GUG showed a much lower relative binding energy (−4.5 kcal/mol) in the absence of both factors, indicating that it may be preferred over AUG in such a scenario.

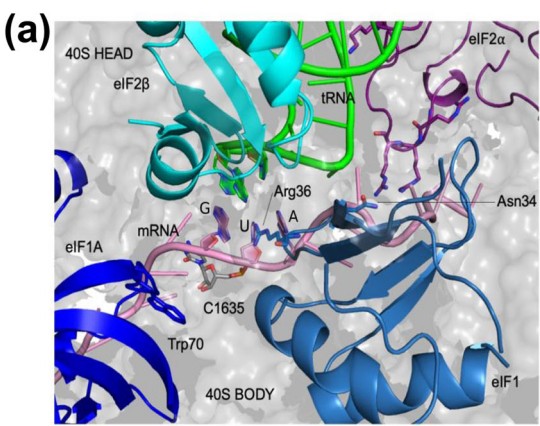
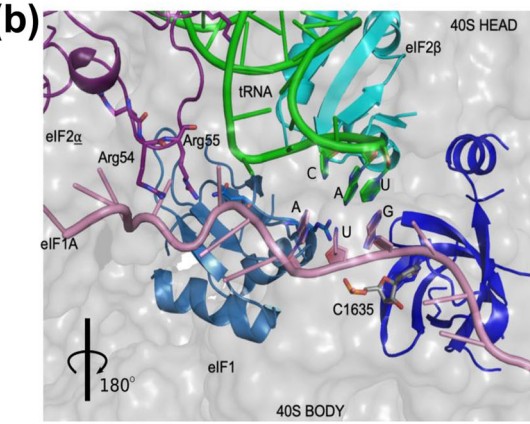

**Fig. 4 Recognition of the start codon AUG by 48S PIC in open conformation. a, b** The key players involved in the codon-anticodon interactions are shown in cartoon representation in two different views rotated by 180°. The corresponding residues which may augment codon: anticodon interactions are shown in stick representations. Arg36 of eIF1 (pale blue), interacts with the first two codon bases, while Trp70 of eIF1A (blue) points towards the mRNA providing stacking interactions. eIF2α shown in purple color contains Arg54 and Arg55 residues (**b**), which interact with mRNA. eIF2β (cyan) interacts with both tRNA$_i$ and eIF1 at the P site. The tRNA$_i$ and mRNA are shown in green and pink, respectively.

Intriguingly, deleting eIF2α from the system conferred a marked reduction in relative binding energy in the case of AUA from ~17.4 to 2.9 kcal/mol, while that of GUG remained virtually unchanged at 1.3 kcal/mol (Fig. 3 and Supplementary Data 2). eIF2α is bound at the E-site and it interacts with both tRNA$_i$ and mRNA upstream of the start codon in the P$_{IN}$ state[20]; however, its role in the fidelity of start codon selection is not well understood. The absence of eIF2α from the system, as discussed later, gives more flexibility to mRNA for base-pairing with the tRNA$_i$ anticodon in the P site (Supplementary Figs. 3 and 4), which might account for the lower energy penalty for the AUA codon.

Eliminating eIF2β results in a small decrease in energy penalty for AUA (from ~17.4 to 11.2 kcal/mol), but an increase for GUG (from 1.1 to ~3 kcal/mol). As described above, for eIF1, these perturbations might be expected to increase initiation at AUA codons but decrease it at GUG or UUG codons. As noted above, mutations expected to weaken eIF2β interaction with eIF1 in the open conformation of the 48S PIC increase UUG initiation[24], which can be explained by increased rearrangement to the closed complex at UUG codons that presumably outweighs the slightly increased discrimination at UUG codons in the open complex predicted from our results in Fig. 3. Considering that eIF2β is also in the vicinity of eIF1A and the tRNA$_i$ anticodon stem-loop in the py48S-open complex[24], the changes in relative binding energy for AUA and GUG in the absence of eIF2β (Fig. 3) might indicate an indirect role in codon selection in the open state by virtue of its interaction with eIF1, eIF1A, or tRNA$_i$.

**Structural insights into AUG codon selection in the scanning conformation of the 48S PIC.** The calculated relative binding energies indicate that the open 48S PIC complex can discriminate against many noncognate as well as near-cognate codons. Interestingly, four near-cognate codons show a low penalty, and the contributions of eIFs to the larger penalty incurred with the AUA triplet were revealed by the reductions in this penalty observed in their absence. Average MD structures taken from the individual runs as well as the extracted structures from the simulation runs were analyzed to figure out the mechanism of codon selection by the 48S PIC in its open conformation. We have extracted and averaged 40 frames of the last 40 ns of the simulation trajectory for each run for a single system to obtain an average MD structure.

The available cryo-EM structures of yeast 48S PIC in an open conformation have an AUC start codon in the mRNA where only the A and U bases of the codon in the P site could be clearly observed[24,25]. Thus, in the absence of a structure of yeast 48S PIC in scanning conformation with an AUG start codon, which has not yet been captured experimentally for structural determination, the average MD structure with an AUG codon provides insights into recognition of the correct start codon in the open state (Fig. 4a, b). This structure reveals a stable codon:anticodon interaction at the P site (Fig. 4a) compared to others. The position of the mRNA in the channel, as well as the codon at the P site in the average MD structure in open state, is different from that observed in the closed state[37] (Supplementary Fig. 5a). Hence, the mRNA including the codon at the P site is repositioned as the mRNA channel is narrowed during the transition from open to the closed state. Interestingly, similar observation in the change of mRNA and start codon position from open to closed conformation of PIC was also observed in bacterial translation initiation[41].

In the average MD structure with AUG codon, Arg36 present in β-hairpin loop-1 of eIF1 interacts with the first two nucleotides (A and U) of the codon (Fig. 4a), whereas this loop of eIF1 has no interaction with the codon bases in the py48S-open and py48S-open-eIF3 structures[24,25]. Notably, the main chain amino group of Arg36 interacts with 'U' of CAU anticodon of the codon:anticodon duplex in py48S and py48S-closed complexes[20,24,25]. Moreover, Asn34, also present in β-hairpin loop-1 of eIF1, interacts with the first nucleotide 'A' of the codon in another average MD structure with AUG codon (Supplementary Fig. 5b); however, in this case, the interaction with Arg36 is not observed. Asn34 interaction with anticodon was seen only in the closed state of 48S PIC[20,24,25]. Thus, interactions observed with Asn34 and Arg36 in the average MD structures seem to indicate how these residues hold onto the codon:anticodon duplex in the open state with wider P site, and these residues continue to interact with codon:anticodon duplex even after the transition to the closed state.

No direct contacts of eIF1A with the mRNA were found in the average MD structure with an AUG codon. Trp70 of eIF1A was observed to stack with +4 nucleotide of the mRNA in py48S and py48S-closed complexes[20,24,25]. However, no interaction between Trp70 and mRNA is observed in the simulation run with AUG (Fig. 4). It is likely that as the mRNA channel is narrowed during the transition from the open state to the closed state, the base of the +4 nucleotide of the mRNA flips out to stack with the Trp70

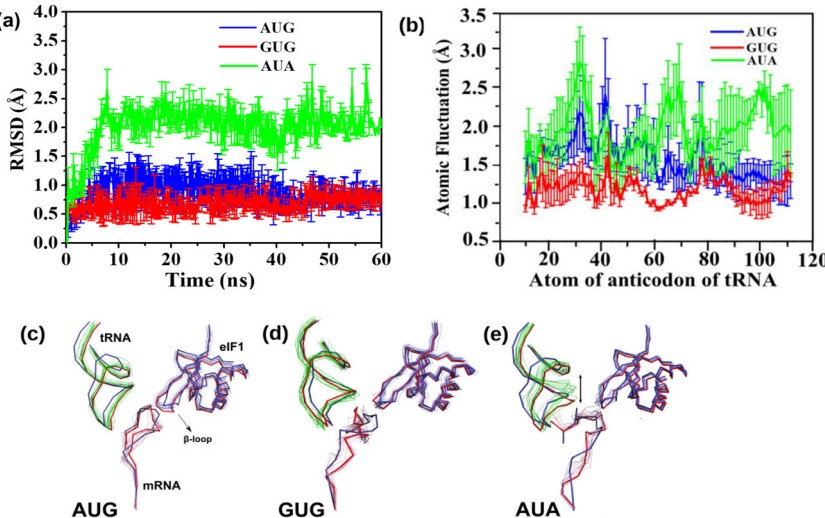

**Fig. 5 Root mean square deviation of tRNA_i backbone and average structures. a** Root mean square deviation (RMSD) of tRNA_i backbone from the independent simulation runs of AUG (blue), AUA (green) and GUG (red), respectively. The error bars indicate the standard error obtained from the mean of four independent simulation runs. Supplementary Data 3 contains the relevant source data. **b** Root mean square fluctuations (RMSF) of atoms in nucleotides of the anticodon of tRNA_i in the case of AUG (blue), GUG (red) and AUA (green) MD simulation run. The fluctuations highlight the dynamic nature of tRNA_i where the nucleotides of tRNA show more fluctuations for AUA (green) when compared to cognate AUG (blue) and near cognate GUG (red) codons. The error bars indicate the standard error obtained from the mean of four independent simulation runs. Supplementary Data 4 contains the relevant source data. **c**–**e** Snapshots of the representative structures extracted from the respective (AUG, GUG and AUA) simulation runs are shown in ribbon representation. The structures are superposed keeping a rRNA stretch from the 40S body as reference. The anticodon stem loop of tRNAi (green) is more stable for **c** cognate and **d** near-cognate codons whereas it is more dynamic for **e** non-cognate codon. The starting and final structures for each individual run are shown in pink and blue, respectively.

of eIF1A. The base of rRNA residue C1635 provides stacking interactions to the third codon base (Fig. 4), which in turn may stabilize the base-pairing between codon:anticodon. C1635 was also observed stacking the third codon base pairing with the anticodon in the 48S PIC structures with AUG codon in the closed state[20,24,25].

Furthermore, Arg residues in a loop of eIF2α could be observed interacting with mRNA upstream of the AUG codon in the P site (Fig. 4). These are similar to the interactions observed in the P_IN state of py48S-closed, but were not observed in the cryo-EM map of py48S-open, probably due to the lack of distinct mRNA density in the widened mRNA channel. Arg54 and Arg55 of eIF2α form hydrogen bonds with the −3 and −4 bases of mRNA (Fig. 4), which belong to the Kozak sequence. Thus, it appears that eIF2α interacts with mRNA and binds to the Kozak sequence upstream of AUG in the P site, even in the open conformation of the 48S PIC. eIF2β is sandwiched between tRNA_i and eIF1 as observed in py48S-open-eIF3 complex and forms hydrogen bonds with both tRNA_i and eIF1. Thus, the average MD structure with an AUG codon provides insights into the interaction of the mRNA including the start codon with eIFs and ribosomal proteins and rRNA in the open state.

**Structural insights into discrimination of non-AUG codons in the scanning conformation of the 48S PIC.** The systems with low energy penalty have a relatively stable tRNA_i during the course of the simulation, which plays a prominent role in forming and maintaining the codon:anticodon interactions. Whereas in scenarios of high energy penalty, the tRNA_i does not stabilize during the course of simulation and the anticodon stem loop (ASL) was observed to be more dynamic (Fig. 5), thus breaking the codon-anticodon interaction. Furthermore, root mean square fluctuations of the tRNA_i revealed that the anticodon residues are relatively more dynamic (Fig. 5b). The β-hairpin loop-1 of eIF1

was found to be dynamic as well. However, apart from loop-1, the overall structure of 'eIF1' was comparatively stable and was superimposable onto one another with a relatively low root mean square deviation score of ~0.8 Å in multiple simulation runs.

The loop-1 makes interactions with the first two nucleotides in the codons in multiple runs with different codons at the P site (Supplementary Fig. 2). Similarly, the interaction of loop-1 with A and U of anticodon is also observed in multiple runs in the open state (Supplementary Fig. 2). However, no interaction with nucleotide at the third position of codon or with corresponding C of anticodon was observed for eIF1. Thus, how eIF1 influences the selection at the third position of the codon and at the same time tolerates mismatch at the first position in the open state is not clear. We suggest that the loop-1 of eIF1 interacting with codon:anticodon in the open state reduces the available space at the P site for any relaxed association between mRNA and tRNA even in widened mRNA channel, thereby discriminating against most codons. Codons with mismatch in first position (GUG, CUG, and UUG) and a codon with a pyrimidine in second position, i.e., ACG, are tolerated even in the presence of eIF1. Further, the third position interaction is stabilized by stacking with C1635 (discussed below) and the presence of loop-1 of eIF1 and rRNA C1635 seems to ensure the selection of the correct base-pair at the third position. In the absence of eIF1 the available space at the P site is increased, thereby decreasing the energy penalty even for third position mismatch (Fig. 3).

In simulations of the open PIC complex containing an AUG start codon, Arg54 and Arg55 of eIF2α interacts with the (−3/−4) bases upstream of the AUG codon in the P site (Figs. 4b and 6a). In the AUA simulation, by contrast, Arg54 and Arg55 appear to shift their position away from mRNA to interact largely with tRNA_i during the course of the simulation (Fig. 6c). Thus, in this scenario where the codon:anticodon interaction is not stable compared to the near cognate ones, these residues can frequently lose their interaction with the mRNA (Fig. 6b, c and

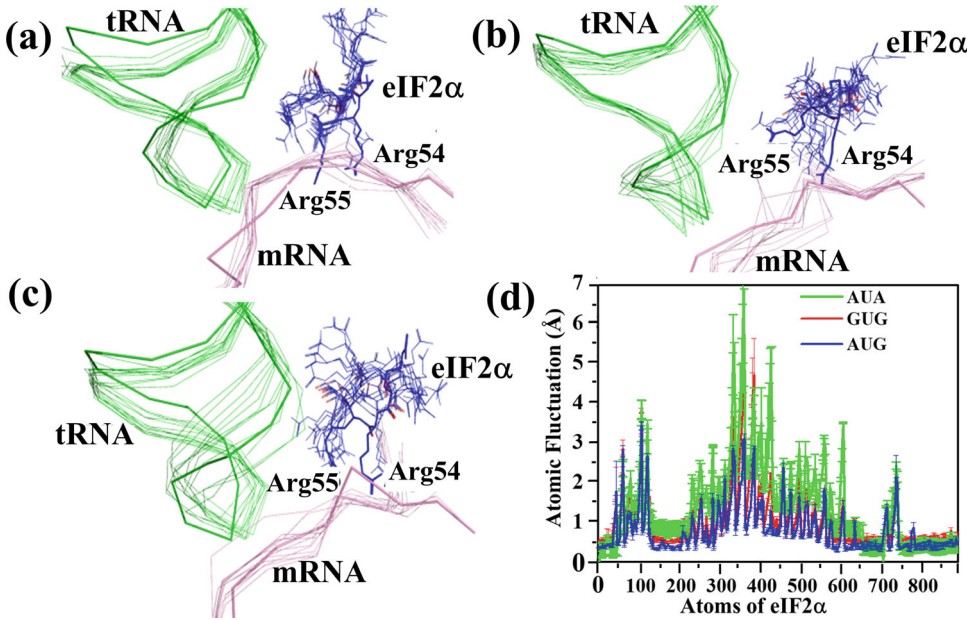

**Fig. 6 Dynamics of Arg54 and Arg55 of eIF2α.** Movement of Arg54 and Arg55 of eIF2α from **a** AUG, **b** GUG and **c** AUA simulation runs are depicted here. The coordinates were extracted from frames of the respective trajectories at 5 ns interval. The starting structures for the runs are shown in a thicker ribbon representation. The movement of mRNA and Arg residues are more prominent for AUA. **d** The root mean square fluctuation (RMSF) of eIF2α atoms. High fluctuation is observed for the loop containing Arg54 and Arg55. The error bars indicate the standard error obtained from the mean of four independent simulation runs. Supplementary Data 5 contains the relevant source data.

Supplementary Fig. 6), which might allow the mRNA to move and bring the next triplet of nucleotides into the P site. Interestingly, in the case of GUG simulations, the movement of eIF2α loop containing Arg54 and Arg55 appears to be less than AUA but more than AUG (Fig. 6d). Thus Arg54 and Arg55 of eIF2α seem to stabilize the mRNA in place in the case of AUG in the widened channel of open conformation of 48S PIC. In the absence of eIF2α from the system the mRNA is found to be more flexible, thereby allowing near-cognate AUA to base-pair with the tRNA$_i$ anticodon in the P site (Supplementary Figs. 3 and 4). Thus, eIF2α may play an important role in the fidelity of codon selection. It would be interesting to check if the absence of eIF2α or mutations of Arg54 and Arg55 increases non-AUG initiation.

eIF2β is positioned between tRNA$_i$ and eIF1 and forms hydrogen bonds with both tRNA$_i$ and eIF1 (Fig. 4). As mentioned above, the eIF2β-eIF1 interaction, found exclusively in py48S-open, appears to increase accuracy by maintaining the scanning conformation of the PIC at near-cognate UUG codons[24,25]. Interestingly, comparing simulations conducted with either AUG, GUG or AUA codons, we observed a moderate decrease in the number of hydrogen bond interactions at the eIF2β-eIF1 interface for GUG versus AUA and a larger reduction for AUG versus both GUG and AUA (Supplementary Fig. 7), which may indicate that the system is shifting towards the closed state upon correct start codon recognition.

The rRNA residue C1635 stacks the third codon base, thereby stabilizing the base-pairing between codon-anticodon in the case of AUG simulations, as discussed above. In the case of GUG simulations, C1635 is also observed to stack the third codon position (Supplementary Fig. 2). Whereas in the case of near-cognate AUA and noncognate codons AGU, ACA, and UAU where no base-pairing is observed between codon-anticodon, C1635 no longer stacks the third base of the codon (Supplementary Fig. 2b). The stacking of C1635 with codon-anticodon is also observed in py48S and py48S-closed complexes[20,24]. Thus, this interaction in AUG and the near cognate GUG codon simulations may indicate that the 48S PIC is preparing for the transition towards the closed conformation upon codon-anticodon base-pairing in the open state.

## Conclusions

Computational studies have been used to bring out the detailed energetics and intermolecular interactions involved in various steps in the process of protein synthesis[42–45]. Earlier, molecular simulations of noncognate and near-cognate tRNA-mRNA interactions were studied in the context of the A site of the bacterial ribosome to evaluate how ribosomes discriminate between correct and incorrect tRNAs during elongation[46]. Recently, MD simulation studies were done to study codon selection in translation initiation in the P site of the closed conformation of the 48S PIC[21,22]. We have carried out MD simulation studies to compare the relative binding energy of the tRNA$_i$ with different noncognate codons with respect to start codon, i.e., AUG in the P site of the open conformation of 48S PIC. A simulation sphere of 40 Å radius centered on of 'A' nucleotide of the anticodon CAU of tRNA$_i$ from py48S-open-eIF3 complex was generated and used for the simulation studies to provide insights into the energetics of start codon selection by the 48S complex in its scanning conformation. Since we study a simulation sphere of 40 Å radius at the P site this study does not provide details of conformational changes in 48S outside the simulation sphere. All-atom simulation runs of 48S PIC would provide a holistic picture of large-scale conformational changes in the 48S during scanning.

The cryo-EM structure of py48S-open-eIF3 complex (PDB ID: 6GSM)[25] does not have densities corresponding to the N- as well as C- terminal tails of eIF1A and hence these are not accounted for in this study. While eIF1A N-terminal tail plays a role in stabilizing the closed conformation of 48 S PIC, the C-terminal tail (CTT) of eIF1A extends into the P site[47] and stabilizes the open conformation of the 48S PIC[48]. Recognition of AUG would lead to the removal of eIF1A-CTT from the P site[49]. In the absence of eIF1A-CTT density in the cryo-EM structure of py48S-open-eIF3 complex[25], this study is also carried out without

taking into account eIF1A-CTT. Hence, how eIF1A-CTT stabilizes $P_{OUT}$ conformation and mutations in CTT facilitate $P_{OUT}$ to $P_{IN}$ transitions at near cognate codon[48] remains to be figured out.

Our MD simulation studies with cognate, near, and noncognate codons in 48S PIC gave insights into the energetics of start codon selection by the PIC in its open state. The result indicates the ability of tRNA$_i$ to preferentially base pair with a cognate start codon (AUG) in $P_{OUT}$ or open state. Base-pairing of the correct codon:anticodon holds the tRNA$_i$ in a widened mRNA channel while β-hairpin loop-1 of eIF1 monitors the codon:anticodon interaction. eIF2α interacts with mRNA at the E site preparing the 48S PIC in open state to change its conformation to closed $P_{IN}$ conformation.

The tRNA dynamics in the widened P site in the open state seem to drive the selection of the codons. Relatively stable codon:anticodon interaction as in the case of AUG and GUG codons compared to other noncognate codons, led to a comparatively stable tRNA during the simulation. The β-hairpin loop-1 of eIF1 protrudes into the P site and interacts with the AUG codon. Though it poses no steric hindrance to the codon:anticodon in the widened P site in the open state, it reduces the available space for an incorrect association between codon and anticodon. However, in the closed state, eIF1 poses a steric hindrance to the complete accommodation of the codon:anticodon interaction in the narrow P site and exerts stricter criteria for recognition of cognate codon:anticodon interaction. This study suggests that eIF1 plays a role in codon selection in the open conformation of 48 S PIC as the simulation runs in the absence of eIF1 leads to a decrease in energy penalty for non-AUG codons.

Moreover, this study also appears to indicate that most codons may be discriminated against by the open form of 48S PIC and only four near-cognate codons (GUG, CUG, UUG and ACG) are likely to pass through to the close form by the measure of energy penalty of binding. Subsets of near-cognate codons have been shown to be the start site for initiation and GUG, CUG, UUG and ACG codons have been reported to initiate translation[1,2,5,27,33,50]. In addition, codons AAG and AGG, which show a high penalty in our study, are also reported not to initiate translation[29,33,50,51]. While AUA, AUU, and AUC codons, which show high penalty in our study, have been shown to act as a start codon in *Neurospora crassa*, their ability to be recognized as alternate start codon is with much lower efficiency[33,51]. Thus, overall the selection of codons by the 48S open form, as suggested by this study, correlates well with the non-AUG codons reported to initiate translation.

Seemingly, the open form of 48S acts as the first step of start codon selection where a coarse selection is done. It would allow only 4 of 63 non-AUG codons to the next closed state for further fine selection. This arrangement of a coarse selection in the open conformation would ensure a high speed of scanning of long 5′ UTR avoiding the need to change conformations to a closed state for each triplet encountered at the P site. Shuttling back and forth between the open and closed states of 48S PIC for every new codon in the P site would require a conformational change in various components of 48S PIC amounting to the making and breaking of several interactions between them. Thus, dynamic switching back and forth between open and closed states may not account for the high speed of scanning by 48S PIC.

The open form of 48S PIC would rule out most of the non-AUG codons, thereby allowing a much more thorough checkup point of only a few near-cognate codons in the closed state. Recognition of correct start codon and accommodation of codon:anticodon in P site in closed state triggers the repositioning and eventually release of eIF1. The vacant site at P site is now occupied by the N-terminal domain of eIF5, which then rechecks the codon in the P site[37]. Thus, allowing the codon:anticodon

interaction to be monitored at multiple checkpoints during initiation ensures a more thorough and robust mechanism of codon selection. This study suggests how the 48S PIC strikes a balance between the accuracy of codon selection and high speed of scanning by employing the open state as a coarse selectivity checkpoint to reject all but a few of the possible codon:anticodon mismatches.

In the future, it would be quite interesting to simulate the whole 48S PIC and look at the functioning mechanism of various initiation factors and elucidate their role in protein translation initiation. Further such studies can help us to understand the mechanism of tRNA selection and the role of Kozak sequences in initiation. These studies can be performed in the closed and open state of PIC which, would provide detailed insights into the initiation process. Thus, MD simulation studies represent an area of great opportunity to understand how these molecular machines work.

## Methods

**Design of models for molecular dynamics simulations**. We based our analysis on the structure of the py48S-open-eIF3 complex (PDB ID: 6GSM) determined at 5.2 Å resolution[25]. Overall this structure is similar to py48S-open complex reported previously (PDB ID: 3JAQ)[24] and has density for three nucleotides (AAU) of mRNA corresponding to 'A' at the −1 position and 'AU' in the +1 and +2 positions of the AUC codon in the P site, and weak density for mRNA observed throughout the mRNA channel. Hence, for the purpose of our calculations, we decided to model the remaining nucleotides of mRNA in the mRNA channel as this would mimic the state when PIC is scanning the 5′ UTR. Since the mRNAs in py48S-open-eIF3 complex (PDB- id: 6GSM) and py48S-eIF5N complex (PDB ID: 6FYY)[37] differ only in a single base (i.e., AUC vs. AUG codon at the P site), we have modeled the mRNA in the remaining mRNA channel based on the mRNA observed in py48S-eIF5N complex (PDB ID: 6FYY).

It is computationally expensive to simulate the whole py48S-open-eIF3 complex, so a simulation sphere of 40 Å radius centered on the center of mass (COM) of the 'A35' nucleotide of the anticodon (5′-CAU-3′) of tRNA$_i$ was generated. In earlier studies, a 25 Å simulation sphere from py48S complex was used for energy calculations of the 48S PIC in the $P_{IN}$ state[21]. Since the $P_{OUT}$ state has a widened mRNA channel, we decided to increase the radius of the simulation sphere to account for this feature, as well as to include more contributions from ribosomal components, tRNA$_i$, and bound eIFs, namely eIF1, eIF1A, eIF2α, and eIF2β.

**Generation of near-cognate and noncognate codons at the P site**. To compare the relative binding energy of the tRNA$_i$ with different noncognate codons in the P site with respect to AUG, 3-, 2-, and 1-point mutations were introduced in the start codon. Thus, the only change in the simulation sphere is in the codon at the P site while the rest of the atomic coordinates remain unchanged. The respective mutations of the start codon at the P site were introduced using the module "mutate_bases" of X3DNA software[52]. In order to study the effect of the eIFs, atomic coordinates of each factor individually, or combinations of eIFs, were excluded from the simulation sphere before the respective production run.

**Molecular dynamics simulation protocol**. Molecular dynamics (MD) simulations were carried out using the PMEMD module of the AMBER14 package[53]. The TIP3P water model[54] was used to solvate such that the solvation shell extends at least 15 Å in all directions from the solute. A requisite number of Na$^+$ ions were added to the systems to maintain overall charge neutrality. The Xleap module of the AMBER14 package was used to solvate and add the ions. AMBER ff14SB force field[55] was used to describe the interactions involving proteins, RNA, and water. Joung-Cheatham ion parameters[56] were used to describe interactions involving ions. Energy minimization was then performed on the solvated systems for 3000 steps using the steepest descent method, followed by 3000 steps of the conjugate gradient method. The atomic coordinates of the solute in the simulation sphere were kept fixed to their initial structure using a harmonic restraint of 500 kcal/mol/Å$^2$ during this initial minimization. This minimization was followed by another conjugate gradient minimization by slowly reducing the force constant of the harmonic restraint on the solute from 20 to 0 kcal/mol/Å$^2$ in five consecutive steps. The system was then gradually heated to a temperature of 300 K in two steps: first, the NVT ensemble was involved in heating from 0 to 100 K in 8 ps and then systems were heated to 300 K at 1 atm pressure using the NPT ensemble in 80 ps. The solute particles were restrained to their initial positions using harmonic restraints with force constant of 20 kcal/mol/Å$^2$ during the whole heating process followed by 500 ps of equilibration run in NPT ensemble using a 2 fs time step for integration. This step was followed by 60 ns of NPT production run where any restraints on the solute were removed, except solute atoms which are beyond 30 Å

from the center of the sphere were harmonically restrained with 10 kcal/mol/Å² force constant. The pressure was kept constant at 1 atm using Berendsen weak coupling method[57].

The short-ranged van der Walls (vdW) and electrostatic interactions were truncated within a real space cut-off of 10 Å and the particle mesh Ewald (PME) method was used to calculate long-range electrostatic interactions. All bond lengths involving hydrogen atoms were constrained using the SHAKE algorithm. Each simulation has been repeated four times with different initial velocities. The snapshots were generated using VMD[58].

**MM-PBSA binding energy calculations.** The MM-PBSA (MM: Molecular Mechanics; PB: Poisson Boltzmann; SA: Surface area) method was used to calculate relative energies of binding using the MMPBSA.py[59] module of AmberTools14. The binding energy ($\Delta E_{bind}$) is expressed as $\Delta E_{bind} = \Delta E_{ele} + \Delta E_{vdw} + \Delta E_{int} + \Delta E_{sol}$, where $\Delta E_{ele}$ is the changes in electrostatic energy, $\Delta E_{vdw}$ is the non-bond van der Waals energy, $\Delta E_{int}$ is the internal energy from bonds, angles, and torsions, and the contribution from the solvent is $\Delta E_{sol}$. $\Delta E_{sol}$ is the sum of the electrostatic energy ($\Delta E_{es}$) and the non-electrostatic energy ($\Delta E_{nes}$). $\Delta E_{es}$ is calculated using the Poisson–Boltzmann (PB) method and the $\Delta E_{nes}$ is expressed as $\Delta E_{nes} = \gamma SASA + \beta$, where $\gamma = 0.00542$ kcal Å$^{-2}$ is the surface tension, $\beta = 0.92$ kcal mol$^{-1}$, and SASA is the solvent-accessible surface area of the molecule. The time series of the binding energy of the codon-anticodon complex was determined using gas-phase energies (MM) and solvation energies following the Poisson Boltzmann model (PB/SA) analysis from snapshots obtained from the last 40 ns of total 60 ns of simulation trajectory and averaged overall the independent runs for each system.

MM/PBSA method is extensively used in the rescoring of binding poses, binding affinity prediction, and virtual screening. The accuracy of the calculated binding energy/free energy depends on a variety of simulation parameters. The parameters such as MD simulation length, choice of the solute dielectric constant, the inclusion of explicit water molecules, and the inclusion of entropy contributions can affect the outcome. Also, previous studies have suggested that instead of a single long simulation, multiple short runs give better binding energy estimates while using MM/PBSA[60]. In our case, we have run multiple short simulations, optimized our system for MM/PBSA parameters, and we expect the calculated binding energies to be comparable to experimental values.

**Reporting summary**. Further information on research design is available in the Nature Research Reporting Summary linked to this article.

## Data availability
The data that support the findings of this study are available from the corresponding author upon request. Supplementary Data 1–5 contain the relevant source data for Figs. 2, 3, 5a, b and 6d, respectively.

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

## Acknowledgements
The authors thank Alan G. Hinnebusch for insightful comments and suggestions. I.B. thanks SERB-National Post Doctoral Fellowship (N-PDF) and B.G. acknowledges Dr. D.S. Kothari Postdoctoral Fellowship (DSKPDF) for financial supports [201718-BL/16-17/0437]. T.C. was supported by a postdoctoral fellowship from the DBT/Wellcome Trust India Alliance grant (IA/I/17/2/503313) awarded to T.H. T.H. acknowledges funding from DBT/Wellcome Trust India Alliance (IA/I/17/2/503313), DST-FIST [SR/FST/LS11-036/2014(C)], UGC-SAP [F.4.13/2018/DRS-III (SAP-II)] and DBT-IISc Partnership Program Phase-II (BT/PR27952-INF/22/212/2018). We thank Sahasrat, SERC, and TUE-CMS, SSCU at IISc, Bangalore, India for the computational facilities. I.B. is also thankful to Dr. Suman Chakrabarty for the GPU computational facilities in the S. N. Bose National Centre for Basic Sciences.

## Author contributions
T.H. and P.K.M. designed research; I.B. and B.G. performed research; I.B., B.G., T.C., and T.H. analyzed data; and I.B., T.C., P.K.M., and T.H. wrote the manuscript.

## Competing interests
The authors declare no competing interests.
