## [Peer Review File · Communications Biology]

Reviewers' comments:

Reviewer #1 (Remarks to the Author):

This manuscript investigates the roles of various initiation factors in a eukaryotic ribosome. The study involves explicit-solvent simulations of a portion of the ribosome (codon-anticodon region) and included APBS calculations to quantify the energetic contributions of each factor to initiator-codon recognition. The strategy is appropriate for the questions posed. In addition, the force field and ion models are suitable for simulations of such a system. Overall, the manuscript is easy to read and clearly presented. However, there are some areas that are in need of improvement.

Issues to address

Major Points

1) There is a lack of quantitative descriptors when discussing structural properties. Rather, most of the structural data plots are given as time traces or structural snapshots. While representative snapshots can be helpful to illustrate a quantitative claim, one can not rely on structural snapshots alone. Here are examples of where quantitative descriptions are necessary: Fig 5 claims that the non-cognate codon is more dynamic. In addition to showing RMSD, which only described deviations and not fluctuations, the authors could report RMSF by residue/atom to quantify the increase in mobility/dynamics. Fig 6 has the same issue. The caption title is "Dynamics of...", but there is not a quantitative description of the dynamics. It does look like panel c may be more mobile, but this may also be an optical illusion (i.e. perspective-dependent). My expectation is that the claims of dynamics will be supported by quantitative measures. But, as presented, the analysis is rather cursory. Same issue in Fig S2. It is not clear whether the authors simply watched movies and then described them in the text, or if these snapshots are truly representative of the simulated dynamics. Fig S3 claims a shift in conformation, but it is not easy to interpret. Specifically, it is very difficult to identify depth in an image that is composed of lines. Fig S5 should also provide averages +/- standard deviations. It is very difficult to tell if the differences between any of the 6 traces are statistically significant.

2) Fig 3 is very difficult to follow. While, given some time, it is possible to understand what quantities are describe, the caption needs improvement. For example, on the left side of the figure there is a "UAC GUG" and on the right it says "UAC AUA", but the caption does not indicate what these labels mean. At first glance, I interpreted these labels to denote the two systems used for calculating relative energies (i.e. difference between UAC and GUG, instead of a UAC-GUG interaction). Since both are using UAC, perhaps "UAC" could be removed from the image. Then, the two sets of data would clearly be intended to compare the GUG vs AUA sequences.

Simple Points:

1) Since the simulations are less than 100ns, it is not possible to make claims about absolute stability. The authors are careful to indicate relative stabilities, when discussing energetic values, but there are cases where forms of the word "stable" are being used to describe interactions. All claims of stability should be rephrased, or removed. That is, if hydrogen bonds are more likely to be formed in one system, then this would suggest the interactions are more stable than in another system. But, having an interaction remain formed for 100ns does not mean it is "stably" formed. It is not uncommon in explicit-solvent simulations of RNA and the ribosome that structural elements may remain formed for more than a microsecond before they rearrange and remain in an alternate conformation indefinitely.

2) Fig 5 title reads "RMSF of tRNAi backbone and average PDBs". The term "average PDB" is not precise. The authors are reporting average structures, calculated from simulations. "PDB" simply refers to a file format. While the authors may have saved their average structure in PDB format, describing them as "average PDBs" is not particularly meaningful.

3) Page 19 reads "the dynamics of these arginines of eIF2 α appears to be intermediate between"

It is not clear how "dynamics" can be "intermediate between". This passage describes Fig 6 and S5, which were mentioned under Major Issues. So, perhaps clarifying the images will also allow the authors to clarify the claim.

4) The conclusions begin with "Computational studies have been used to bring out the detailed energetics and intermolecular interactions involved in various steps in the process of protein synthesis^{51,52,53,54,55,56,57}". Ref 51 is a 2ns targeted MD simulation that can not provide any insights into energetics or contact formation. Essentially, the dynamics are entirely determined a priori by the targeting constraint. Accordingly, this reference should be removed from this block. If the authors want to mention this study, due to its historical significance (i.e. first simulation of a ribosome with explicit solvent), it would be acceptable to present it in that light. Reference 57 should be removed. All other references in this block are research articles that provide analysis of energetics. 57 is simply a review of large-scale simulations, without a clear focus on energetics in the ribosome.

5) In the conclusions, the text reads "Our extensive MD simulation studies with cognate, near and non-cognate codons in 48S PIC gave crucial insights into the energetics of start codon selection by the PIC in its open state." In relation to the current field, these simulations are not considered "extensive", even if they provide a reasonably clear account of the system. Removing "extensive" would be appropriate. Next, it is recommended that the authors do not try to over-sell the significance of the results. It would be advisable to remove "crucial" and let the field decide how significant the insights truly are.

Reviewer #2 (Remarks to the Author):

Basu and co-workers present an exciting study of start codon selection by the eukaryotic ribosome during mRNA scanning. This is a key process not only for basic understanding of protein synthesis in eukaryotes, but also in terms of mRNA vaccine design. The authors perform extensive molecular simulations for a variety of matched and mismatched codon-anticodon interactions, investigating which mismatches are more stable than others and how the initial factors contribute to the stability of the codon-anticodon-ribosome hydrogen bond network. The work is of high impact and, with significant revision, important to publish in Nature Communications Biology.

Specific comments:

1. While the authors present a thorough review of previous work in eukaryotic systems, they should place their work in the context of bacterial ribosome to appeal to a more general audience. This is the area where the vast majority of mechanistic work has been performed. More specifically, the author's study is similar to a previous study in the bacterial ribosome on tRNA selection, where molecular simulations of a large number of non-cognate and near-cognate tRNA-mRNA interactions were studied in the context of the ribosome to evaluate the effect of mismatches on the stability of the codon-anticodon-ribosome hydrogen bond network (Sanbonmatsu and Joseph, JMB, 2003). The authors should also use bacterial rRNA number in parentheses, in addition to the eukaryotic numbering.
2. The authors simulate a small region of the ribosome (a 40 Å sphere around the region of interest) and make claims about the full, intact ribosome complex. It is possible to perform explicit solvent simulations of the entire ribosome complex with comparable or more sampling than the current submission (several such studies have been published since 2010, including Whitford, et al., RNA 2010; Whitford, et al., JACS 2010; Whitford et al., PLoS Comp. Biol. 2013). While performing simulations of the full 80S ribosome complex for every case may be beyond the scope of this study, the authors should perform simulations of the 80S complex for at least one case to help validate the approximation that their 40 Å sphere is representative of the full complex.
3. In relation to comment #2 (above), the authors do consider multiple conformations of the complex in the sense that they simulate the open and closed conformations; however, the role of large-scale fluctuations between open to closed and to open again should be addressed in the

discussion. These motions are critical for making the transition between the two states. Thus, it would be useful to discuss the barrier height (Munro, et al., Biopolymers 2008) and the role of kinetics vs. thermodynamics in scanning and selection.

4. In terms of the interpretation of the data, this was thorough, but would benefit greatly if the authors make more specific predictions for specific future experiments, based on their simulations.

5. To calculate free energies of binding, the authors use the molecular mechanics Poisson Boltzmann surface area (MM-PBSA). This method does not explicitly include the entropic component of the free energy, makes an implicit solvent approximation based on Poisson Boltzmann and instead uses an ad-hoc correction factor (e.g., "Assessing the performance of MM/PBSA and MM/GBSA methods: Entropy effects on the performance of end-point binding free energy calculation approaches", Phys. Chem. Chem. Phys., Royal Soc. Chem., 2018). More accurate methods use enhanced sampling to explicitly sample conformational space and thus include the entropic component in the free energy calculation (e.g., $\Delta G = -kT \log(P)$). To validate the MM-PBSA results, the authors should employ an enhanced sampling method for at least one case (e.g., Hamiltonian replica exchange molecular dynamics or metadynamics) and also discuss the limitations of the MM-PBSA estimate.

6. More discussion should be included on how the initial starting structures were chosen for the near-cognate and non-cognate cases, since no experimental data exists regarding the majority of these. Importantly, how can the authors be sure that shifts in energies are not the result of initial large changes due to the construction of the initial starting model, which includes interactions not present in experimental structural data. The authors should also discuss how the near-cognate tRNAs could, in the living cell, arrive at the starting configurations used in the simulations and whether or not there are likely additional proof reading steps involved in the initial, non-specific binding process that occurs before the system reaches the specific interactions in the bound form.

Reviewer #3 (Remarks to the Author):

In their manuscript "Selection of start codon during mRNA scanning in eukaryotic translation initiation" Basu et al. use molecular dynamics simulations to address the question of how the preinitiation complex (PIC) selects the mRNA start codon in eukaryotes.

For different codons, they compare codon-anticodon binding energies obtained from MD simulations of the open PIC conformation. Further, they removed several initiation factors to monitor their effect on the binding energies and therefore their role in selection.

The addressed questions are very important to understand initiation of translation and therefore of interest to a broad audience. However, based on the presented results, I am not convinced yet that the conclusions are sufficiently justified.

As detailed below, my most important concern is the insufficient discussion of the convergence of the simulations.

Major points:

The authors base a lot of their conclusions on average MD structures (e.g, Fig. 4), but it is not explained how these structures were obtained. Were all frames of all replicates averaged?

More importantly, it is not clear how similar the structural ensembles obtained from the different replicates are, i.e. how converged the trajectories are. Here, for example, the rmsd of the codon-anticodon base pairs from the respective starting structures as a function of simulation time would help to quantify how converged the simulations are.

It is not clear what the presented structures in Figure S2 are. Average structures? Conformations extracted from the simulations? If yes, based on what criteria?

How different do the conformations look for different replicates?

Fig S5: What criterion did the authors use to decide that the interaction for AUG and GUG are stable but not for AUA? The number of hydrogen bonds is wildly fluctuating for all codons.

The simulations are based on the py48S-open-eIF3 complex which shows that the bases A and U

of the AUC codon base pair with the tRNA anticodon nucleotides. However, as shown in Figure S2, in the simulations with the AUC codon the tRNA moves far away from the mRNA which is not compatible with the cryo-EM structure. What might be the reason for that? Further, if this conformation is not captured correctly, how can the authors be sure that the simulations of the other codons capture what would happen in the PIC?

On page 12, the authors write:

"Moreover, the singlepoint transversion mutations at the second position (U) to either purine (i.e. AUG→AAG and AUG→AGG) are not tolerated and likewise confer large relative binding energies, which was not observed for the transition mutation of U→C in the AUC triplet (Fig. 2b)"
I do not follow this conclusion, because the values for AAG, AGG, and AUC are similar and for sure within mutual error bars.

Minor points:

The terms "transition mutation" and "transversion mutation" are used in the manuscript. For a broad audience, these terms should be introduced.

Fig. 2: Since there are only four data points for each codon, drawing each data point as a dot on top of the bar would be helpful to judge the convergence.

Fig. 4: If I am not mistaken, the labeling of the codon-anticodon nucleotides in panel (b) have to be reversed, because the view is rotated by ~180 degrees (?).

Fig. S2 the caption does not match the figure. E.g., (a) should refer to (C)...

Several times the authors refer to "extracted PDBs". This is misleading, because PDB is the Protein Data Bank. The authors should write "extracted conformations", "coordinates" or "structures" instead.

Page 21: the abbreviation NTT is used, but was not introduced before. At least I did not find it.

Some spotted typos:

Abstract: ... 5'UTRs that *are* inspected ...

Page 8: Joung-Cheatham ion parameters *were* used ..

Response to the reviewers' comments on the manuscript COMMSBIO-21-1535

Comments from Reviewer #1 (Remarks to the Author):

General comment: *This manuscript investigates the roles of various initiation factors in a eukaryotic ribosome. The study involves explicit-solvent simulations of a portion of the ribosome (codon-anticodon region) and included APBS calculations to quantify the energetic contributions of each factor to initiator-codon recognition. The strategy is appropriate for the questions posed. In addition, the force field and ion models are suitable for simulations of such a system. Overall, the manuscript is easy to read and clearly presented. However, there are some areas that are in need of improvement.*

Author's comment: We thank for the reviewer for suggestions to improve this manuscript.

Issues to address

Major Points

Comment #1.1: *There is a lack of quantitative descriptors when discussing structural properties. Rather, most of the structural data plots are given as time traces or structural snapshots. While representative snapshots can be helpful to illustrate a quantitative claim, one cannot rely on structural snapshots alone. Here are examples of where quantitative descriptions are necessary: Fig 5 claims that the non-cognate codon is more dynamic. In addition to showing RMSD, which only described deviations and not fluctuations, the authors could report RMSF by residue/atom to quantify the increase in mobility/dynamics.*

Author's comment: We thank for the reviewer for the suggestion. We have incorporated the RMSF by atom of the anticodon to highlight the dynamic nature of tRNA as a new Fig. 5b.

RMSF of atoms in nucleotides of the anticodon of tRNA_i in the case of AUG (Black lines), GUG (Red lines) and AUA (Green lines) MD simulation run. RMSF by atom of the anticodon to highlight the dynamic nature of tRNA_i. The residues of tRNA show more

fluctuations for AUA (represented in green) when compared to cognate AUG (represented in black) and near cognate GUG (represented in red) codons.

Comment #1.2: Fig 6 has the same issue. The caption title is "Dynamics of...", but there is not a quantitative description of the dynamics. It does look like panel c may be more mobile, but this may also be an optical illusion (i.e. perspective-dependent). My expectation is that the claims of dynamics will be supported by quantitative measures. But, as presented, the analysis is rather cursory.

Author's comment: We have removed the word 'Dynamics' from the figure legend and made appropriate modifications. Further we have modified Figure 6 a-c to make it more clear. Also, we have now supported the interpretation of more mobility in panel c (i.e. AUA run) by calculating the root mean square fluctuation (RMSF) of eIF2 α loop that contains conserved Arg54 and Arg55 in AUG (blue line), GUG (red line) and AUA (green line) runs in a new Fig. 6d.

In order to further quantify the movement of Arg54 & Arg55 residues of eIF2 α we calculated the distance between the said residues and mRNA in a new supplementary figure S5. The following figure indicates that the distance between Arg residues of eIF2 α and mRNA is stabilize for the AUG run but not for the AUA run.

(a)

(b)

Comment #1.3: *Same issue in Fig S2. It is not clear whether the authors simply watched movies and then described them in the text, or if these snapshots are truly representative of the simulated dynamics.*

Authors reply: Thanks to the referee for the comment. We wanted to show the binding mode of the codon-anticodon pairs here. So, we have extracted 40 frames of the last 40ns of the simulation trajectory for each run for single system and the snapshot of the averaged structures are represented in the figure S2. We have modified the figure legend to incorporate this information.

Comment #1.4: *Fig S3 claims a shift in conformation, but it is not easy to interpret. Specifically, it is very difficult to identify depth in an image that is composed of lines.*

Authors reply: We agree with the reviewer and have modified this figure (now figure S3a) and placed a P site tRNA as a reference and highlighted the shift by an arrow. Further, in order to have a better representation of the shift we have calculated the distance between the codon-anticodon nucleotides (new Fig S3b). In a new Supplementary figure S3b it can be clearly visualized that in the absence of eIF2 α , AUA seems to get the necessary space to have stable codon-anticodon interactions (red lines).

The plot represents distance in Å between the centre of mass of codon and the centre of mass of anticodon nucleotides of the AUA MD simulation trajectory in the presence (black lines) and in the absence of eIF2α (red lines). Panel 1-3 represents the snapshots (with larger observed fluctuations) extracted at 1) t=24 ns; 2) t=43 ns; 3) t=58 ns of AUA MD simulation trajectory in the presence of eIF2α and panel 4-6 represents the snapshots extracted at the same time line from MD simulation run of AUA in the absence of eIF2α.

Comment #1.5: Fig S5 should also provide averages +/- standard deviations. It is very difficult to tell if the differences between any of the 6 traces are statistically significant.

Authors reply: Thanks to the referee for pointing this out. We agree that it is difficult to tell the difference from this graph. So, we have replaced this with a new supplementary figure S5 and have made necessary changes in the manuscript. In the new supplementary figure S5 we quantify the movement of Arg (54-55) residues of eIF2α by calculating the distance between the said residues and mRNA (in place of H-bonds between Arg54 & Arg55 and mRNA). The figure indicates that the distance between Arg residues of eIF2α and mRNA decreases and

stabilize for AUG but not for GUG and AUA runs. The AUA run shows the highest distance between Arg residues and mRNA.

Comment #2: Fig 3 is very difficult to follow. While, given some time, it is possible to understand what quantities are describe, the caption needs improvement. For example, on the left side of the figure there is a "UAC GUG" and on the right it says "UAC AUA", but the caption does not indicate what these labels mean. At first glance, I interpreted these labels to denote the two systems used for calculating relative energies (i.e. difference between UAC and GUG, instead of a UAC-GUG interaction). Since both are using UAC, perhaps "UAC" could be removed from the image. Then, the two sets of data would clearly be intended to compare the GUG vs AUA sequences.

Authors reply: We thank the reviewer for this suggestion and we have modified the figure as suggested in the revised manuscript.

Simple Points:

Comment #1: Since the simulations are less than 100ns, it is not possible to make claims about absolute stability. The authors are careful to indicate relative stabilities, when discussing energetic values, but there are cases where forms of the word "stable" are being used to describe interactions. All claims of stability should be rephrased, or removed. That is, if hydrogen bonds are more likely to be formed in one system, then this would suggest the interactions are more stable than in another system. But, having an interaction remain formed for 100ns does not mean it is "stably" formed. It is not uncommon in explicit-solvent simulations of RNA and the ribosome that structural elements may remain formed for more than a microsecond before they rearrange and remain in an alternate conformation indefinitely.

Authors Reply: Thanks to the reviewer for the comment. We have made necessary changes as follows:

Pg 18-19: The base of rRNA residue C1635 provides stacking interactions to the third codon base (Fig. 4), which in turn **may** stabilize the base-pairing between codon:anticodon.

Pg 19: Arg54 and Arg55 of eIF2 α form **stable** hydrogen bonds with the -3 and -4 bases of mRNA (Fig. 4), which belong to the Kozak sequence.

Pg 19: The systems with low energy penalty have a **relatively** stable tRNA_i during the course of the simulation, which plays a prominent role in forming and maintaining the codon:anticodon interactions.

Pg 19-20: However, apart from loop-1, the overall structure of 'eIF1' was **comparatively** stable....

Pg 20: In simulations of the open PIC complex containing an AUG start codon, Arg54 and Arg55 of eIF2 α **stably** interacts with the (-3/-4) bases upstream of the AUG codon in the P site (Figs. 4b and 6a).

Pg 21: eIF2 β is positioned between tRNA_i and eIF1 and forms **favorable-stable** hydrogen bonds with both tRNA_i and eIF1 (Fig. 4).

Comment #2: *Fig 5 title reads "RMSF of tRNA_i backbone and average PDBs". The term "average PDB" is not precise. The authors are reporting average structures, calculated from simulations. "PDB" simply refers to a file format. While the authors may have saved their average structure in PDB format, describing them as "average PDBs" is not particularly meaningful.*

Authors reply: We thank the reviewer for pointing this out. The word 'PDB' have been replaced by 'structure' throughout in the revised manuscript.

Comment #3: *Page 19 reads "the dynamics of these arginines of eIF2 α appears to be intermediate between" It is not clear how "dynamics" can be "intermediate between". This passage describes Fig 6 and S5, which were mentioned under Major Issues. So, perhaps clarifying the images will also allow the authors to clarify the claim.*

Authors reply: Thanks to the reviewer for pointing this out. We have modified the statement to mention about the movement of arginine residues as reflected by RMSF of eIF2 α loop that contains conserved Arg54 and Arg55.

“Interestingly, in the case of GUG simulations, the movement of eIF2 α loop containing Arg54 and Arg55 appears to be less than AUA but more than AUG (Fig. 6d).”

Further, we have responded to the comment about Fig 6 and S5 earlier.

Comment #4: *The conclusions begin with "Computational studies have been used to bring out the detailed energetics and intermolecular interactions involved in various steps in the process of protein synthesis^{51,52,53,54,55,56,57}". Ref 51 is a 2ns targeted MD simulation that can not provide any insights into energetics or contact formation. Essentially, the dynamics are entirely determined a priori by the targeting constraint. Accordingly, this reference should be removed from this block. If the authors want to mention this study, due to its historical significance (i.e. first simulation of a ribosome with explicit solvent), it would be acceptable to present it in that light. Reference 57 should be removed. All other references in this block are research articles that provide analysis of energetics. 57 is simply a review of large-scale simulations, without a clear focus on energetics in the ribosome.*

Authors Reply: We thank the reviewer for this suggestion. We agree and references 51 and 57 have been removed.

Comment #5: *In the conclusions, the text reads "Our extensive MD simulation studies with cognate, near and non-cognate codons in 48S PIC gave crucial insights into the energetics of start codon selection by the PIC in its open state." In relation to the current field, these simulations are not considered "extensive", even if they provide a reasonably clear account of the system. Removing "extensive" would be appropriate. Next, it is recommended that the authors do not try to over-sell the significance of the results. It would be advisable to remove "crucial" and let the field decide how significant the insights truly are.*

Authors reply: We are thankful for the suggestion. We have made changes throughout the manuscript as per the suggestion in the revised manuscript. The details of the changes made are mentioned below.

Pg 22: We have carried out **extensive** MD simulations studies to compare the relative binding energy of the tRNA_i with different noncognate codons....

Pg 22: A simulation sphere of 40 Å radius centered on of 'A' nucleotide of the anticodon CAU of tRNA_i from py48S-open-eIF3 complex was generated and used for the simulation studies to provide **crucial** insights into the energetics of start codon selection by the 48S complex in its scanning conformation.

Pg 23: Our **extensive** MD simulation studies with cognate, near and non-cognate codons in 48S PIC gave **crucial** insights into the energetics of start codon selection by the PIC in its open state.

Pg 23: Base-pairing of the correct codon:anticodon **stabilizes** holds the tRNA_i in a widened mRNA channel while β -hairpin loop-1 of eIF1 monitors the codon:anticodon interaction.

Pg 23: **Relatively** stable codon:anticodon interaction as in case of AUG and GUG codons compare to other non-cognate codons, lead to a **comparatively** stable tRNA during the simulation.

Reviewer #2 (Remarks to the Author):

General comment: *Basu and co-workers present an exciting study of start codon selection by the eukaryotic ribosome during mRNA scanning. This is a key process not only for basic understanding of protein synthesis in eukaryotes, but also in terms of mRNA vaccine design. The authors perform extensive molecular simulations for a variety of matched and mismatched codon-anticodon interactions, investigating which mismatches are more stable than others and how the initial factors contribute to the stability of the codon-anticodon-ribosome hydrogen bond network. The work is of high impact and, with significant revision, important to publish in Nature Communications Biology.*

Authors Reply: We appreciate the reviewer's positive remark.

Specific comments:

Comment #1: *While the authors present a thorough review of previous work in eukaryotic systems, they should place their work in the context of bacterial ribosome to appeal to a more general audience. This is the area where the vast majority of mechanistic work has been performed. More specifically, the author's study is similar to a previous study in the bacterial ribosome on tRNA selection, where molecular simulations of a large number of non-cognate and near-cognate tRNA-mRNA interactions were studied in the context of the ribosome to evaluate the effect of mismatches on the stability of the codon-anticodon-ribosome hydrogen bond network (Sanbonmatsu and Joseph, JMB, 2003). The authors should also use bacterial rRNA number in parentheses, in addition to the eukaryotic numbering.*

Authors Reply: In our study we have studied the codon selection at the P site during the process of scanning of 5' UTR. This step is missing in bacterial translation. In bacteria, the base-pairing of Shine Dalgarno (SD) sequence with the 3' end of 16S rRNA places the AUG codon at the P site. Hence, previous studies (including *Sanbonmatsu and Joseph, JMB, 2003*) in bacterial ribosome on tRNA selection was done looking at tRNA selection at A site during elongation step. So, in these case, the codon in mRNA remains while the tRNA is selected for at the A site. Whereas in our study, the initiator tRNA remains constant whereas the mRNA codon changes at the P site (not A site). Hence it is difficult to place this study in context of bacterial ribosome.

However, we have added a sentence in Discussion on Pg 22 to mention the previous work on tRNA selection in bacterial ribosomes.

“Earlier, molecular simulations of non-cognate and near-cognate tRNA-mRNA interactions were studied in the context of the A site of bacterial ribosome to evaluate how ribosomes discriminate between correct and in correct tRNAs during elongation⁵⁶.”

Comment #2: *The authors simulate a small region of the ribosome (a 40 Å sphere around the region of interest) and make claims about the full, intact ribosome complex. It is possible to perform explicit solvent simulations of the entire ribosome complex with comparable or more sampling than the current submission (several such studies have been published since 2010, including Whitford, et al., RNA 2010; Whitford, et al., JACS 2010; Whitford et al., PLoS Comp. Biol. 2013). While performing simulations of the full 80S ribosome complex for every case may be beyond the scope of this study, the authors should perform simulations of the 80S complex for at least one case to help validate the approximation that their 40 Å sphere is representative of the full complex.*

Authors Reply: We thank the reviewer for this suggestion. In the current study we have attempted to understand the process of scanning and selection of start codon in the context of 48S PIC. We agree that the simulation of the whole 48S complex will provide more detailed insights into the initiation process, but unfortunately due to the paucity of computational resources currently, we are unable to do all-atom simulation of full 48S PIC. We have now mentioned this limitation of the study on Pg 22.

“Since we study a simulation sphere of 40 Å radius at the P site this study does not provide details of conformational changes in 48S outside the simulation sphere. All-atom simulation runs of 48S PIC would provide a wholistic picture of large-scale conformational changes in the 48S during scanning.”

In future we will try to study the whole 48S PIC. However, this study provides insights into how scanning of 5' UTR is achieved at a high speed accurately and the results obtained here corroborate with the existing information of usage of non-AUG codons in initiation in eukaryotes.

Comment #3: *In relation to comment #2 (above), the authors do consider multiple conformations of the complex in the sense that they simulate the open and closed conformations; however, the role of large-scale fluctuations between open to closed and to open again should be addressed in the discussion. These motions are critical for making the transition between the two states. Thus, it would be useful to discuss the barrier height (Munro, et al., Biopolymers 2008) and the role of kinetics vs. thermodynamics in scanning and selection.*

Authors Reply: Thank you for this suggestions. It would have been interesting to explore the large-scale fluctuations of open to closed conformations. However, in the case of our study

we are only looking at the codon selection in the a simulation sphere of 40 Å radius at the P site of the open conformation of 48S PIC during the scanning process. As mentioned earlier, this is pointed out in discussion on Pg 22:

“A simulation sphere of 40 Å radius centered on of ‘A’ nucleotide of the anticodon CAU of tRNA_i from py48S-open-eIF3 complex was generated and used for the simulation studies to provide insights into the energetics of start codon selection by the 48S complex in its scanning conformation. Since we study a simulation sphere of 40 Å radius at the P site this study does not provide details of conformational changes in 48S outside the simulation sphere. All-atom simulation runs of 48S PIC would provide a wholistic picture of large-scale conformational changes in the 48S during scanning.”

We’ll attempt performing all-atom simulation runs in future.

Further, we have already discussed on Pg 24 how this study indicate that the open state may be sufficient to discriminate a majority of the non-cognate codons and hence large-scale conformational changes from open to close to open again would not be needed for a majority of non-AUG codons.

“Seemingly, the open form of 48S acts as first step of start codon selection where a coarse selection is done. It would allow only 4 of 63 non-AUG codons to the next closed state for further fine selection. This arrangement of a coarse selection in the open conformation would ensure high speed of scanning of long 5’ UTR avoiding the need to change conformations to closed state for each triplet encountered at the P site. Shuttling back and forth between the open and closed states of 48S PIC, for every new codon in the P site would require conformational change in various components of 48S PIC amounting to making and breaking of several interactions between them. In lieu of which, it can be observed that dynamic switching back and forth between open and closed states may not account for the high speed of scanning by 48S PIC.”

Comment #4: *In terms of the interpretation of the data, this was thorough, but would benefit greatly if the authors make more specific predictions for specific future experiments, based on their simulations.*

Authors Reply: As mentioned above, the results obtained in this study corroborate with the existing information of usage of non-AUG codons in initiation in eukaryotes. Interestingly, simulation runs in the absence of eIF2α showed lower penalty for noncognate AUA codon.

We have added 1-2 sentences on Pg 21 predicting more initiation at non-AUG codons in a system without eIF2 α . Further it would be interesting to check if a similar effect is observed in case of mutation of conserved Arg54 and Arg55 of eIF2 α .

“In the absence of eIF2 α from the system the mRNA is found to be more flexible, thereby allowing near-cognate AUA to base-pair with the tRNA_i anticodon in the P site (Supplementary Fig. S3). Thus, eIF2 α may play an important role in the fidelity of codon selection. It would be interesting to check if absence of eIF2 α or mutations of Arg 54 and Arg55 increases non-AUG initiation.”

Comment #5: *To calculate free energies of binding, the authors use the molecular mechanics Poisson Boltzmann surface area (MM-PBSA). This method does not explicitly include the entropic component of the free energy, makes an implicit solvent approximation based on Poisson Boltzmann and instead uses an ad-hoc correction factor (e.g., “Assessing the performance of MM/PBSA and MM/GBSA methods: Entropy effects on the performance of end-point binding free energy calculation approaches”, Phys. Chem. Chem. Phys., Royal Soc. Chem., 2018). More accurate methods use enhanced sampling to explicitly sample conformational space and thus include the entropic component in the free energy calculation (e.g., $\Delta G = -kT \log(P)$). To validate the MM-PBSA results, the authors should employ an enhanced sampling method for at least one case (e.g., Hamiltonian replica exchange molecular dynamics or metadynamics) and also discuss the limitations of the MM-PBSA estimate.*

Authors Reply: We agree with the reviewer and have discussed the limitations of MM_PBSA estimates on Pg 11.

“MM/PBSA method is extensively used in the rescoring of binding poses, binding affinity prediction, and in virtual screening. The accuracy of the calculated binding energy/free energy depends on a variety of simulation parameters. The parameters such as MD simulation length, choice of solute dielectric constant, inclusion of explicit water molecules, and the inclusion of entropy contributions can affect the outcome. Also, previous studies have suggested that instead of single long simulation, multiple short runs give better binding energy estimates while using MM/PBSA³⁵. In our case, we have run multiple short simulations, optimised our system for MM/PBSA parameters and believe the free energy scores reported here will be relatively identical to the experimental free energies.”

Further, we incorporated the entropy contribution using Quasi-Harmonic approximation in MM-PBSA calculation for a AUG, UUG and GUG. Inclusion of entropy in binding energy also follows similar trend as before when we have for only the enthalpic contribution. This shows that our conclusions about the trend of binding energies remain the similar. Hence we have not included it in the manuscript.

The results of entropy calculations are shown below for your reference:

Entropy calculated using Quasi-harmonic approximation (best two runs):

Codon	Runs	ΔE_{bind} (Kcal/mol)	$T\Delta S_{\text{bind}}$ (Kcal/mol)	ΔG_{bind} (Kcal/mol)
AUG	Run1	-20.24±3.36	-12.45	-7.79
	Run2	-19.7±2.96	-12.27	-7.43
GUG	Run1	-18.17±4.49	-13.01	-5.16
	Run2	-17.28±3.02	-12.51	-4.77
UUG	Run1	-16.73±3.73	-11.56	-5.17
	Run2	-17.60±3.26	-11.77	-5.83

(Relative binding energies for GUG and UUG are similar with/without entropy calculations.)

We do agree that more calculations will cement our reporting but unfortunately, at the moment we have paucity of computational and human resources.

Comment #6: *More discussion should be included on how the initial starting structures were chosen for the near-cognate and non-cognate cases, since no experimental data exists regarding the majority of these. Importantly, how can the authors be sure that shifts in energies are not the result of initial large changes due to the construction of the initial starting model, which includes interactions not present in experimental structural data.*

The authors should also discuss how the near-cognate tRNAs could, in the living cell, arrive at the starting configurations used in the simulations and whether or not there are likely additional proof reading steps involved in the initial, non-specific binding process that occurs before the system reaches the specific interactions in the bound form.

Authors reply: We have mentioned how the initial structure was chosen in the methods section on Pg 8. Briefly, we choose the structure of py48S-open-eIF3 complex (PDB ID: 6GSM) which has density for three nucleotides (AAU) of mRNA corresponding to ‘A’ at the -1 position and ‘AU’ in the +1 and +2 positions of the AUC codon in the P site, and weak density for mRNA observed throughout the mRNA channel. Hence, for the purpose of our calculations, we decided to model the remaining nucleotides of mRNA in the mRNA channel as this would mimic the state when PIC is scanning the 5’ UTR . We have modelled the mRNA in the remaining mRNA channel based on the mRNA observed in py48S-eIF5N

complex (PDB ID: 6FYY). A simulation sphere of 40 Å radius centered on the center of mass (COM) of the 'A35' nucleotide of the anticodon (5'-CAU-3') of tRNA_i was generated which include contributions from ribosomal components, tRNA_i and bound eIFs, namely eIF1, eIF1A, eIF2 α and eIF2 β .

To compare the relative binding energy of the tRNA_i with different noncognate codons in the P site with respect to AUG, 3-, 2- and 1- point mutations were made in the start codon using the module "mutate_bases" of X3DNA software. Hence, the initial starting structures for AUG and mutant codons were generated identically. The only difference was in the mutated base. thus, the only change in the simulation sphere is in the codon at the P site while the rest of the atomic coordinates remain unchanged. So, there is no large difference between the codons in the starting structure except the difference in the start codon. Further, we study only relative binding energies in our study, hence we reason that any shifts in energies is because of the difference in codon (thereby difference in binding energy) and not the result of different initial starting models.

Regarding how the *'near-cognate tRNAs could, in the living cell, arrive at the starting configurations used in the simulations and whether or not there are likely additional proof reading steps involved in the initial, non-specific binding process that occurs before the system reaches the specific interactions in the bound form'*; the case does not apply to our study. This is the case of selection of tRNA at the A site. Whereas in our study, we study the selection of codons in the presence of initiator tRNA at the P site. Here the mRNA is scanned and it moves to bring in new codons at the P site. Hence the two processes are very different. There are additional check-point for codon selection like the closed state, which is more critical for AUG selection. The additional check-points for codon selection (i.e. closed state of 48S PIC) is mentioned on Pg 3, 6-7 (Introduction) and on Pg 25 (Discussion).

Pg 3: "Upon recognition of the start codon the PIC undergoes conformational changes to form a scanning-arrested closed (P_{IN}) complex accompanied by the release of eIF1, which is essential for the fidelity of start codon selection, and dissociation of P_i²."

Pg 6-7: "Our studies also indicate that eIF1 plays a crucial role in codon selectivity in the open conformation of the 48S PIC. However, recognition of AUG as start codon is still inaccurate owing to the failure to discriminate against codons (GUG, CUG, UUG) with a first base-pair codon:anticodon mismatch. Hence, the open conformation of the 48S PIC serves as an initial checkpoint for selection of the start codon at which almost all non-cognate codons are rejected. A

few near-cognate codons, which are accepted in the open state can then be re-examined in a more stringent, second checkpoint, i.e. the closed conformation of the 48S PIC.”

Pg 25: “The open form of 48S PIC would rule out most of the non-AUG codons, thereby allowing a much more thorough checkup point of only a few near-cognate codons in the closed state. Recognition of correct start codon and accommodation of codon:anticodon in P site in closed state triggers the repositioning and eventually release of eIF1. The vacant site at P site is now occupied by the N-terminal domain of eIF5, which then rechecks the codon in the P site²⁶. Thus, allowing the codon:anticodon interaction to be monitored at multiple checkpoints during initiation ensures a more thorough and robust mechanism of codon selection.”

Reviewer #3 (Remarks to the Author):

General comment: *In their manuscript “Selection of start codon during mRNA scanning in eukaryotic translation initiation” Basu et al. use molecular dynamics simulations to address the question of how the preinitiation complex (PIC) selects the mRNA start codon in eukaryotes. For different codons, they compare codon-anticodon binding energies obtained from MD simulations of the open PIC conformation. Further, they removed several initiation factors to monitor their effect on the binding energies and therefore their role in selection.*

The addressed questions are very important to understand initiation of translation and therefore of interest to a broad audience. However, based on the presented results, I am not convinced yet that the conclusions are sufficiently justified. As detailed below, my most important concern is the insufficient discussion of the convergence of the simulations.

Authors Reply: The authors gratefully acknowledge the constructive comments provided by the reviewers.

Convergence of simulation is important issue. To address this, we have run multiple short simulations instead of running single long simulation. Because of the large system size, running the simulation longer than 50-100 ns is computationally challenging. Below we show RMSD of tRNA (Figure 5a) as a function of simulation time to demonstrate the convergence. The RMSD shows that within 60 ns it converged.

Major points:

Comment #1: *The authors base a lot of their conclusions on average MD structures (e.g, Fig. 4), but it is not explained how these structures were obtained. Were all frames of all replicates averaged?*

Authors Reply: Thanks to the referee for this point. We have extracted and averaged 40 frames of the last 40ns of the simulation trajectory for each run for a single system to obtain an average MD structure. It is now mentioned on Pg 17 in the manuscript.

“We have extracted and averaged 40 frames of the last 40 ns of the simulation trajectory for each run for a single system to obtain an average MD structure.”

Comment #2: *More importantly, it is not clear how similar the structural ensembles obtained from the different replicates are, i.e. how converged the trajectories are. Here, for example, the rmsd of the codon-anticodon base pairs from the respective starting structures as a function of simulation time would help to quantify how converged the simulations are.*

Author’s response: Please see our response on convergence in the previous page.

We also show in Fig S3b the time evolution of the centre of mass of codon with the centre of mass of anticodon nucleotides of the mRNA in AUA run in presence of all eIFs (red lines, Fig S3b) and we see the convergence with time.

The structural ensembles obtained from the different replicates are similar. Please see the response to comment 3 below.

Comment #3: *It is not clear what the presented structures in Figure S2 are. Average structures? Conformations extracted from the simulations? If yes, based on what criteria? How different do the conformations look for different replicates?*

Authors Reply: Thanks to the referee for pointing this out. Structures from the last 40ns simulation skipping every 1ns were extracted for each codon were averaged and the average structure is presented in the Figure S2 to show the difference between the binding modes of the different codons. This information is now mentioned in the figure legend. The overall trend of ‘base-pairing or not’ is conserved in different replicates. Below we show average structures of two runs of GUG.

Comment #4: *Fig S5: What criterion did the authors use to decide that the interaction for AUG and GUG are stable but not for AUA? The number of hydrogen bonds is wildly fluctuating for all codons.*

Author's response: We agree that from older figure S5, it was not clearly evident. We have now replaced this figure with a new supplementary figure S5 and have made necessary changes in the manuscript. In the new supplementary figure S5 we quantify the movement of Arg (54-55) residues of eIF2 α by calculating the distance between the said residues and mRNA (in place of H-bonds between Arg54 & Arg55 and mRNA). The figure indicates that the distance between Arg residues of eIF2 α and mRNA decreases and stabilize for AUG but tend to increase for GUG and AUA runs. So, the interaction is stable for AUG only. We have modified the manuscript on Pg 20-21 to reflect the same.

Comment #4: *The simulations are based on the py48S-open-eIF3 complex which shows that the bases A and U of the AUC codon base pair with the tRNA anticodon nucleotides. However, as shown in Figure S2, in the simulations with the AUC codon the tRNA moves far away from the mRNA which is not compatible with the cryo-EM structure. What might be the reason for that? Further, if this conformation is not captured correctly, how can the authors be sure that the simulations of the other codons capture what would happen in the PIC?*

Authors reply: The initial structures for the simulation runs were obtained from PDB 6GSM, solved using cryo-EM, where the authors were able to capture only a tiny fraction of the particles in this conformation. This indicates it is quite tricky to capture AUC codon in open state of 48S PIC. In our study, the starting structures of AUC codons were identical to those obtained from the PDB 6GSM, but gradually the tRNA starts to shift away from mRNA which may explain why it is difficult to capture AUC in open state as evident by capturing only a tiny fraction of the particles in this conformation in the cryo-EM study.

We had a re-look at all AUC runs and we observe that codon:anticodon do not base pair in AUC runs. However, we figured out a mistake in the energy penalty for AUC runs. The average relative binding energy is about 7.6 kcal/mol instead of about 12 kcal/mol. We have corrected this in Fig 2b and made corresponding changes in the manuscript. However, this does not change the overall interpretation of the results. It still show a higher penalty than near-cognate ACG, CUG, UUG and GUG.

Overall, the results obtained here corroborate with the existing information of usage of non-AUG codons in initiation in eukaryotes. Hence, we reason that this study in spite of its

limitations the study does provide insights into codon selection at high speed during scanning process.

Comment #5: *On page 12, the authors write: “Moreover, the singlepoint transversion mutations at the second position (U) to either purine (i.e. AUG→AAG and AUG→AGG) are not tolerated and likewise confer large relative binding energies, which was not observed for the transition mutation of U→C in the AUC triplet (Fig. 2b)” I do not follow this conclusion, because the values for AAG, AGG, and AUC are similar and for sure within mutual error bars.*

Authors Reply: Thanks to the referee for pointing this out. This should be ACG instead of AUC. U to C mutation of AUG is ACG, not AUC, and we can see the energetic penalty in case of ACG is much lower than that of AAG and AGG. This is modified at page 14 in the revised manuscript.

“Moreover, the single-point transversion mutations at the second position (U) to either purine (i.e. AUG→AAG and AUG→AGG) are not tolerated and likewise confer large relative binding energies, which was not observed for the transition mutation of U→C in the ACG triplet (Fig. 2b).”

Minor points:

Comment #1: *The terms “transition mutation” and “transversion mutation” are used in the manuscript. For a broad audience, these terms should be introduced.*

Authors Reply: As per the suggestion we have introduced the terms on Pg 12.

“Out of twenty-seven possible triplets with mutations from AUG at all three positions (3-point mutations), relative binding energies were calculated for two such triplets selected at random, ensuring that that both transition (point mutations involving the interchanges of two ring purine base i.e. A↔G, or of one ring pyrimidine base i.e. C↔U;)) and transversion (point mutations involving change from purine to pyrimidine base or *vice versa*) mutations at all three positions were represented.”

Comment #2: *Fig. 2: Since there are only four data points for each codon, drawing each data point as a dot on top of the bar would be helpful to judge the convergence.*

Authors Reply: Please see our response about convergence earlier. Also, we now provide binding energies of all runs (depicted in Figs. 2 & 3) in two supplementary Tables (Tables S1 and S2).

Comment #3: *Fig. 4: If I am not mistaken, the labeling of the codon-anticodon nucleotides in panel (b) have to be reversed, because the view is rotated by ~180 degrees (?).*

Authors reply: The figure has been updated appropriately. The labelling in panel (a) is corrected.

Comment #4: *Fig. S2 the caption does not match the figure. E.g., (a) should refer to (C)...*

Authors reply: Thanks to the referee for pointing this out. Yes there is a mistake while putting the caption. It is modified in the revised manuscript.

Comment #5: *Several times the authors refer to “extracted PDBs”. This is misleading, because PDB is the Protein Data Bank. The authors should write “extracted conformations”, “coordinates” or “structures” instead.*

Authors reply: Thanks for pointing this out. We have changed ‘PDBs’ to ‘structures’ in the revised manuscript.

Comment #6: *Page 21: the abbreviation NTT is used, but was not introduced before. At least I did not find it.*

Authors reply: The abbreviation of NTT (N-terminal tail) was already introduced on Pg 5 but for the benefit of the readers we have expanded it again on Pg 22 (earlier Pg 21).

Pg 22: While eIF1A N-terminal tail plays a role in stabilizing the closed conformation of 48S PIC,...

Comment #3: Some spotted typos:

Abstract: ... 5’UTRs that *are* inspected ... Page 8: Joung-Cheatham ion parameters *were* used ..

Authors Reply: Thanks to the referee for pointing this out. We have corrected them in the revised manuscript.

Reviewers' comments:

Reviewer #1 (Remarks to the Author):

The revised manuscript is significantly improved. The authors present a clear question: Is there a physical basis for the ribosome to scan start codons in the open conformation, or is a transition to the close conformation necessary. Through comparative energetic analysis of numerous system, the authors provide evidence that a significant level of discrimination can be imparted by the open conformation. The manuscript is well written, overall. There are some additional comments that the authors should consider, in order to more clearly present their findings.

1) There are many minor typos that should be cleaned up. I counted at least 15.

2) I think "the structure of partial yeast 48S complex" should be "a partial structure of the yeast 48S complex"

3) "Furthermore, the simulation of coordinates of a closed-state 48S" should probably be "Furthermore, the simulation that were initialized from the closed-state of 48S..."

4) comment: "This led us to consider whether the 48S PIC in an open conformation can accurately recognize the start codon and discriminate against noncognate codons while scanning the 5' UTR. If this is indeed true, then it may provide insights into codon selection during scanning and also an explanation for the high speed of scanning as the ribosomal initiation complex would not have to undergo a large conformational change (from open to closed state and back) to inspect every single incoming nucleotide triplet in the P site."

I just wanted to note that it is excellent that the authors clearly presented a question. It is (unfortunately) very common for simulation studies to have no clear objective, but rather just aim to provide broad descriptions of a system, which often relegated simulations to the role of advanced movie-making. Properly stating the question already makes this study superior to many simulations studies that have been published.

5) "Thus, our study provides novel insights into how the 48S maintains accuracy at a high rate of scanning by utilising the open state as a coarse selectivity checkpoint to reject all but a few of the possible codon:anticodon mismatches." This is a slight overstatement, since the results have not been corroborated experimentally. It should be rephrased to say "into how the 48S can maintain accuracy..." The fact that the results have not yet been corroborated is not a shortcoming of the study, in my opinion. But, it can be damaging to the integrity of the field to (unintentionally) imply that the simulated results are more than predictions (even if they are very likely to be correct).

6) "It is computationally expensive to simulate the whole py48S-open-eIF3 complex, so a simulation sphere of 40 Å radius centred on the centre of mass (COM) of the 'A35' nucleotide of the anticodon (5'-CAU-3') of tRNA_i was generated." This is a weak justification for the methods. Simply claiming insufficient computing resources does not make the applied method suitable. A more appropriate description would be to state the assumptions when approaching the problem in this way. For example, by using a subset of atoms, one is assuming that long-range electrostatics and possible long-range "allosteric" effects are not dominant factors that control codon discrimination. I think those assumptions are reasonable, since the quantities of interest are relative energetic values.

Related to this point, I noticed that another reviewer suggested the results need to be validated through comparison with full-ribosome simulations. Unfortunately, to the best of my knowledge, there has yet to be any full-ribosome simulations that have been rigorously validated by experiments. Certainly various all-atom explicit-solvent simulations from Bock and Grumbuller, or Whitford and Sanbonmatsu have been able to suggest physical properties of the ribosome. But, even their full-ribosome simulations have been quite short, with only one extending to multiple microseconds. Given the size of the ribosome, it is still not clear if current force fields will ensure that the structure of the full assembly is stable. Even if the force field have very large inaccuracies, global reorganization processes are likely to require much more time than has been accessible. It

is also possible that the divalent ion models are insufficient to ensure stability of the ribosome. Based on the difficulties of properly simulating small RNA molecules (e.g. Chen and Garcia, PNAS 2013), there is a lot of room for minor inaccuracies to lead to large-scale effects on the ribosome, though it may take many microseconds to see the aggregate effect of these errors. Perhaps with the development of the Anton III supercomputer, and recently-refined force fields by the Shaw group, it will be possible to see if the ribosome is stable on accessible simulated timescales.

7) "In our case, we have run multiple short simulations, optimised our system for MM/PBSA parameters and believe the free energy scores reported here will be relatively identical to the experimental free energies." First, I am not sure what a "free energy score" is. "score" is used again to describe RMSD. Second, "identical" should not be used, since nothing is ever identical (except mathematical expression). Third, one should probably avoid the term "believe", since that sounds more like faith than science. All three points could be addressed by rephrasing the text as "we expect the calculated free energies to be comparable to experimental values."

8) With regards to updating the references, the reply said that reference 57 was removed, but the review article in question was actually just moved to reference 55. I had to dig through my files to see what happened. It appears the authors may have accidentally removed the original number 58, instead (Yu, Y., et al. Position of eukaryotic translation initiation factor eIF1A on the 40S ribosomal subunit mapped by directed hydroxyl radical probing. *Nucleic Acids Res.* 37, 5167–5182 (2009).).

9) With regards to improving the quantitative evidence supporting the claims, the new figures make the points much clearer. However, the plots should have a uniform appearance. There are several plots that have extremely small axis labels (e.g. Fig. 6d). As far as label sizes, all figures should be comparable to 5b. Also, the labels should not overlap with axes.

10) The caption to figure 5 refers to black lines. I assume this should be blue.

Reviewer #2 (Remarks to the Author):

1. I am still concerned about the reviewer's comment:

The authors simulate a small region of the ribosome (a 40 Å sphere around the region of interest) and make claims about the full, intact ribosome complex. It is possible to perform explicit solvent simulations of the entire ribosome complex with comparable or more sampling than the current submission (several such studies have been published since 2010, including Whitford, et al., RNA 2010; Whitford, et al., JACS 2010; Whitford et al., PLoS Comp. Biol. 2013). While performing simulations of the full 80S ribosome complex for every case may be beyond the scope of this study, the authors should perform simulations of the 80S complex for at least one case to help validate the approximation that their 40 Å sphere is representative of the full complex.

and the author's response:

We thank the reviewer for this suggestion. In the current study we have attempted to understand the process of scanning and selection of start codon in the context of 48S PIC. We agree that the simulation of the whole 48S complex will provide more detailed insights into the initiation process, but unfortunately due to the paucity of computational resources currently, we are unable to do all-atom simulation of full 48S PIC. We have now mentioned this limitation of the study on Pg 22. "Since we study a simulation sphere of 40 Å radius at the P site this study does not provide details of conformational changes in 48S outside the simulation sphere. All-atom simulation runs of 48S PIC would provide a wholistic picture of large-scale conformational changes in the 48S during scanning." In future we will try to study the whole 48S PIC.

The authors need to perform some simulations of the whole 48S system to provide some evidence that there are no major changes between the full system and the localized system and that their

major conclusions will still hold. This is an essential control.

2. Regarding the reviewer's comment:

To calculate free energies of binding, the authors use the molecular mechanics Poisson Boltzmann surface area (MM-PBSA). This method does not explicitly include the entropic component of the free energy, makes an implicit solvent approximation based on Poisson Boltzmann and instead uses an ad-hoc correction factor (e.g., "Assessing the performance of MM/PBSA and MM/GBSA methods: Entropy effects on the performance of end-point binding free energy calculation approaches", Phys. Chem. Chem. Phys., Royal Soc. Chem., 2018). More accurate methods use enhanced sampling to explicitly sample conformational space and thus include the entropic component in the free energy calculation (e.g., $\Delta G = -kT \log(P)$). To validate the MM-PBSA results, the authors should employ an enhanced sampling method for at least one case (e.g., Hamiltonian replica exchange molecular dynamics or metadynamics) and also discuss the limitations of the MM-PBSA estimate.

The authors response was inadequate:

"MM/PBSA method is extensively used in the rescoring of binding poses, binding affinity prediction, and in virtual screening. The accuracy of the calculated binding energy/free energy depends on a variety of simulation parameters. The parameters such as MD simulation length, choice of solute dielectric constant, inclusion of explicit water molecules, and the inclusion of entropy contributions can affect the outcome. Also, previous studies have suggested that instead of single long simulation, multiple short runs give better binding energy estimates while using MM/PBSA. In our case, we have run multiple short simulations, optimised our system for MM/PBSA parameters and believe the free energy scores reported here will be relatively identical to the experimental free energies."

Further, we incorporated the entropy contribution using Quasi-Harmonic approximation in MM-PBSA calculation for a AUG, UUG and GUG. Inclusion of entropy in binding energy also follows similar trend as before when we have for only the enthalpic contribution. This shows that our conclusions about the trend of binding energies remain the similar. Hence we have not included it in the manuscript.

It is well known that MM-PBSA including the quasi-harmonic approximation cannot produce accurate free energies. It is well known that this technique has inadequate treatment of the entropic contributions. For example, Nobel Laureate Ari Warshel states "This approach appears to provide erroneous estimates of the absolute binding energies due to its incorrect entropies and the problematic treatment of electrostatic energies." (Absolute Binding Free Energy Calculations: On the Accuracy of Computational Scoring of Protein-ligand Interactions, Singh and Warshel, Proteins, 2010). To validate the MM-PBSA results, the authors should employ an enhanced sampling method such as Hamiltonian replica exchange molecular dynamics, metadynamics, or umbrella sampling.

3. Upon close inspection of the manuscript, I am concerned about how the authors obtained their average structures. They state:

"So, we have extracted 40 frames of the last 40ns of the simulation trajectory for each run for single system and the snapshot of the averaged structures are represented in the figure S2."
and

"We have extracted and averaged 40 frames of the last 40ns of the simulation trajectory for each run for a single system to obtain an average MD structure. It is now mentioned on Pg 17 in the manuscript."

Simple averaging of structure coordinates can yield sterically unfeasible models. One proper way to do this would be to make histograms (or energy landscapes) and take a typical structure from the dominant basin, rather than averaging structure coordinates.

Reviewer #3 (Remarks to the Author):

The manuscript has improved a lot. In my opinion, all of the issues, except for the convergence issue, were sufficiently addressed by the authors.

In their reply, the authors claim that convergence is reached within 60 ns based on the rmsd plot (Fig. 5a). However, the rmsd of single trajectories does not suffice to test for convergence. For example, if the authors would have stopped their simulations after 8 ns, the rmsds would also look rather flat, because the increase in rmsd for the AUA simulation happens around 10 ns. Therefore, with the same reasoning, the authors could have claimed based on the 8 ns simulations that they were converged. This would obviously not have been true given the increase in rmsd that is seen at 10 ns with longer simulations. It would be more convincing if they see similar levels of rmsd reached in the repetitions. I do not understand why replicates are not shown.

Another argument for convergence presented by the authors in the reply is that the average structures of two runs of the GUG simulations are similar. Showing only these two examples is not convincing, because it is not clear why these were selected. Is this similarity observed for all repetitions of each codon?

It is not yet clear to me how many simulations were run for each codons, in the methods part, the authors write that simulations were repeated four to six times, but in all the time traces (Figs. 5, S5, and S6) only one trace per codon is shown. How were they selected?

To improve transparency, the authors should provide a list of all simulations (Supplement) and also show the time traces for all simulations.

Reviewer #1 (Remarks to the Author):

The revised manuscript is significantly improved. The authors present a clear question: Is there a physical basis for the ribosome to scan start codons in the open conformation, or is a transition to the close conformation necessary. Through comparative energetic analysis of numerous system, the authors provide evidence that a significant level of discrimination can be imparted by the open conformation. The manuscript is well written, overall. There are some additional comments that the authors should consider, in order to more clearly present their findings.

1) There are many minor typos that should be cleaned up. I counted at least 15.

Authors reply: We are thankful to the reviewer for pointing this out. We have checked for the typos and corrected them.

2) I think "the structure of partial yeast 48S complex" should be "a partial structure of the yeast 48S complex".

Authors reply: We are thankful to the reviewer for pointing this out. We have modified the sentence in the manuscript accordingly.

3) "Furthermore, the simulation of coordinates of a closed-state 48S" should probably be "Furthermore, the simulation that were initialized from the closed-state of 48S..."

Authors reply: We have modified the sentence in the manuscript as suggested.

4) comment: "This led us to consider whether the 48S PIC in an open conformation can accurately recognize the start codon and discriminate against noncognate codons while scanning the 5' UTR. If this is indeed true, then it may provide insights into codon selection during scanning and also an explanation for the high speed of scanning as the ribosomal initiation complex would not have to undergo a large conformational change (from open to closed state and back) to inspect every single incoming nucleotide triplet in the P site."

I just wanted to note that it is excellent that the authors clearly presented a question. It is (unfortunately) very common for simulation studies to have no clear objective, but rather just aim to provide broad descriptions of a system, which often relegated simulations to the role of advanced movie-making. Properly stating the question already makes this study superior to many simulation studies that have been published.

Authors reply: We are thankful to the reviewer for this encouraging comment.

5) "Thus, our study provides novel insights into how the 48S maintains accuracy at a high rate of scanning by utilising the open state as a coarse selectivity checkpoint to reject all but a few of the possible codon:anticodon mismatches." This is a slight overstatement, since the results have not been corroborated experimentally. It should be rephrased to say "into how the 48S can maintain accuracy..." The fact that the results have not yet been corroborated is not a shortcoming of the study, in my opinion. But, it can be damaging to the integrity of the field to (unintentionally) imply that the simulated results are more than predictions (even if they are very likely to be correct).

Authors reply: We have modified the sentence on page 7 in the manuscript.

6) "It is computationally expensive to simulate the whole py48S-open-eIF3 complex, so a simulation sphere of 40 Å radius centred on the centre of mass (COM) of the 'A35' nucleotide of the anticodon (5'-CAU-3') of tRNA_i was generated." This is a weak justification for the methods. Simply claiming insufficient computing resources does not make the applied method suitable. A more appropriate description would be to state the assumptions when approaching the problem in this way. For example, by using a subset of atoms, one is assuming that long-range electrostatics and possible long-range "allosteric" effects are not dominant factors that control codon discrimination. I think those assumptions are reasonable, since the quantities of interest are relative energetic values.

Related to this point, I noticed that another reviewer suggested the results need to be validated through comparison with full-ribosome simulations. Unfortunately, to the best of my knowledge, there has yet to be any full-ribosome simulations that have been rigorously validated by experiments. Certainly various all-atom explicit-solvent simulations from Bock and Grumbuller, or Whitford and Sanbonmatsu have been able to suggest physical properties of the ribosome. But, even their full-ribosome simulations have been quite short, with only one extending to multiple microseconds. Given the size of the ribosome, it is still not clear if current force fields will ensure that the structure of the full assembly is stable. Even if the force field have very large inaccuracies, global reorganization processes are likely to require much more time than has been accessible. It is also possible that the divalent ion models are insufficient to ensure stability of the ribosome. Based on the difficulties of properly simulating small RNA molecules (e.g. Chen and Garcia, PNAS 2013), there is a lot of room for minor inaccuracies to lead to large-scale effects on the ribosome, though it may take many microseconds to see the aggregate effect of these errors. Perhaps with the development of the Anton III supercomputer, and recently-refined force fields by the Shaw group, it will be possible to see if the ribosome is stable on accessible simulated timescales.

Authors reply: We are thankful to the reviewer for this elaborate comment. We have calculated relative energy values by simulating a sphere around the codon:anticodon interactions. While doing so we have ensured that important factors that may contribute significantly to relative binding values are within the simulation sphere. And long-range 'allosteric' effects are unlikely to play a dominant role in codon discrimination as tRNA binds to the 40S head and this interactions remains unchanged during large conformational changes during the initiation pathways (Llacer JL and Hussain T et al, Mol Cell, 2015). Simulations spheres around codon:anticodon have been used to study codon:anticodon pairs earlier (Lind et al. Nucleic Acids Res 2016 and Sanbonmatsu, K. Y. et al. J. Mol. Biol. 2003).

7) "In our case, we have run multiple short simulations, optimised our system for MM/PBSA parameters and believe the free energy scores reported here will be relatively identical to the experimental free energies." First, I am not sure what a "free energy score" is. "score" is used again to describe RMSD. Second, "identical" should not be used, since nothing is ever identical (except mathematical expression). Third, one should probably avoid the term "believe", since that sounds more like faith than science. All three points could be addressed by rephrasing the text as "we expect the calculated free energies to be comparable to experimental values."

Authors reply: We have modified the sentence on page 11 in the manuscript.

8) *With regards to updating the references, the reply said that reference 57 was removed, but the review article in question was actually just moved to reference 55. I had to dig through my files to see what happened. It appears the authors may have accidentally removed the original number 58, instead (Yu, Y., et al. Position of eukaryotic translation initiation factor eIF1A on the 40S ribosomal subunit mapped by directed hydroxyl radical probing. Nucleic Acids Res. 37, 5167–5182 (2009).).*

Authors reply: We apologize for the error and thank the reviewer for pointing this out. This has been corrected in the revised manuscript.

9) *With regards to improving the quantitative evidence supporting the claims, the new figures make the points much clearer. However, the plots should have a uniform appearance. There are several plots that have extremely small axis labels (e.g. Fig. 6d). As far as label sizes, all figures should be comparable to 5b. Also, the labels should not overlap with axes.*

Authors reply: The figures have been modified accordingly.

10) *The caption to figure 5 refers to black lines. I assume this should be blue.*

Authors reply: The captions are corrected now.

Reviewer #2 (Remarks to the Author):

1. I am still concerned about the reviewer's comment:

The authors simulate a small region of the ribosome (a 40 Å sphere around the region of interest) and make claims about the full, intact ribosome complex. It is possible to perform explicit solvent simulations of the entire ribosome complex with comparable or more sampling than the current submission (several such studies have been published since 2010, including Whitford, et al., RNA 2010; Whitford, et al., JACS 2010; Whitford et al., PLoS Comp. Biol. 2013). While performing simulations of the full 80S ribosome complex for every case may be beyond the scope of this study, the authors should perform simulations of the 80S complex for at least one case to help validate the approximation that their 40 Å sphere is representative of the full complex.

and the author's response:

We thank the reviewer for this suggestion. In the current study we have attempted to understand the process of scanning and selection of start codon in the context of 48S PIC. We agree that the simulation of the whole 48S complex will provide more detailed insights into the initiation process, but unfortunately due to the paucity of computational resources currently, we are unable to do all-atom simulation of full 48S PIC. We have now mentioned this limitation of the study on Pg 22. "Since we study a simulation sphere of 40 Å radius at the P site this study does not provide details of conformational changes in 48S outside the simulation sphere. All-atom simulation runs of 48S PIC would provide a wholistic picture of large-scale conformational changes in the 48S during scanning." In future we will try to study the whole 48S PIC.

The authors need to perform some simulations of the whole 48S system to provide some evidence that there are no major changes between the full system and the localized system and that their major conclusions will still hold. This is an essential control.

Authors reply: While large-scale conformational changes are clearly important in overall initiation pathway, it is ultimately the codon:anticodon interaction at the P site of the 40S which determines the codon selection during translation initiation. Hence, we have calculated relative energy values

by simulating a sphere around the codon:anticodon interactions and while doing so we have ensured that eIFs that may contribute to relative binding values are within the simulation sphere. And long-range 'allosteric' effects are unlikely to play a dominant role in codon discrimination as tRNA binds to the 40S head and this interactions remains unchanged during large conformational changes during the initiation pathways (Llacer JL and Hussain T et al, Mol Cell, 2015). Moreover, simulations spheres around codon:anticodon have been used to study codon:anticodon pairs earlier (Lind et al. Nucleic Acids Res 2016 and Sanbonmatsu, K. Y. et al. J. Mol. Biol. 2003). Further, the selection of codons by the 48S open form, as suggested by this study correlates well with the non-AUG codons reported to initiate translation. Thus our approach is justified by above mentioned reasons.

However, 'to provide some evidence that there are no major changes between the full system and the localized system and that our major conclusions will still hold' we have run one MD simulation with the whole ribosome of the P-OUT structure. After solvation and addition of ions, the whole system comprised of ~2500000 atoms. We have run one single simulation of 200ns long. (Write details of simulation run, like force-field etc, etc). The TIP3P water model was used to solvate. Requisite number of Na⁺ ions were added to the systems to maintain overall charge neutrality. The Xleap module of the AMBER14 package was used to solvate and add the ions. AMBER ff14SB force field was used to describe the interactions involving proteins, RNA, and water. Joung-Cheatham ion parameters were used to describe interactions involving ions.

Figure 1: Representation of the whole structure.

To observe the stability of the codon-anticodon interaction in the whole ribosome, we have calculated the distance between the center of mass (COM) of AUG codon with the center of mass (COM) of CAU anticodon of the tRNA and plotted the probability distribution of the distance (shown in figure below). This shows a strong single peak around 1.1 nm which indicates that the strong stability of codon-anticodon pairing throughout the simulation time. Thus codon-anticodon interaction remains stable in the whole ribosome just like in our truncated ribosome (having AUG as start codon) throughout the simulation run. The most populated cluster of the whole ribosome is represented to show the stable interaction between the codon and anticodon in the figure (2b).

Figure 2: (a) Probability distribution of the distance between the center of mass (COM) of the codon and anticodon in the whole ribosome. (b) The representative snapshot corresponds to the most populated cluster also shows the stability of the interaction.

This whole ribosome simulation result is in agreement with the truncated simulation of ribosome having AUG codon. In both cases, the codon-anticodon interaction is stable throughout the simulation time. This strongly indicates that our truncated ribosome simulation results are meaningful.

Study	System	System Size (in atoms)	Simulation Run time
Whitford et al., PLoS Comp. Biol. 2013	Fully solvated 70S ribosome	2070120	1300 ns
Whitford et al. J. Am. Chem. Soc. 2010, 132, 38, 13170–13171	Fully solvated 70S ribosome	~3200000	7 runs (200 ns – 300 ns each)
Whitford et al., RNA 2010, 16(6), 1196–1204	Fully solvated 70S ribosome	3178833	17 runs (ranging from 12 ns to 204 ns)
Our study	Fully solvated 48S PIC	2453200	200 ns

Figure 3: RMSD vs time of tRNA in the whole ribosome simulation. This indicates the stability of the tRNA in favourable codon-anticodon interaction.

2. Regarding the reviewer's comment:

To calculate free energies of binding, the authors use the molecular mechanics Poisson Boltzmann surface area (MM-PBSA). This method does not explicitly include the entropic component of the free energy, makes an implicit solvent approximation based on Poisson Boltzmann and instead uses an ad-hoc correction factor (e.g., “Assessing the performance of MM/PBSA and MM/GBSA methods: Entropy effects on the performance of end-point binding free energy calculation approaches”, Phys. Chem. Chem. Phys., Royal Soc. Chem., 2018). More accurate methods use enhanced sampling to explicitly sample conformational space and thus include the entropic component in the free energy calculation (e.g., $\Delta G = -kT \log(P)$). To validate the MM-PBSA results, the authors should employ an enhanced sampling method for at least one case (e.g., Hamiltonian replica exchange molecular dynamics or metadynamics) and also discuss the limitations of the MM-PBSA estimate.

The authors response was inadequate:

“MM/PBSA method is extensively used in the rescoring of binding poses, binding affinity prediction, and in virtual screening. The accuracy of the calculated binding energy/free energy depends on a variety of simulation parameters. The parameters such as MD simulation length, choice of solute dielectric constant, inclusion of explicit water molecules, and the inclusion of entropy contributions can affect the outcome. Also, previous studies have suggested that instead of single long simulation, multiple short runs give better binding energy estimates while using MM/PBSA. In our case, we have run multiple short simulations, optimised our system for MM/PBSA parameters and believe the free energy scores reported here will be relatively identical to the experimental free energies.”

Further, we incorporated the entropy contribution using Quasi-Harmonic approximation in MM-PBSA calculation for a AUG, UUG and GUG. Inclusion of entropy in binding energy also follows similar trend as before when we have for only the enthalpic contribution. This shows that our conclusions about the trend of binding energies remain the similar. Hence we have not included it in the manuscript.

It is well known that MM-PBSA including the quasi-harmonic approximation cannot produce accurate free energies. It is well known that this technique has inadequate treatment of the entropic contributions. For example, Nobel Laureate Ari Warshel states “This approach appears to provide erroneous estimates of the absolute binding energies due to its incorrect entropies and the problematic treatment of electrostatic energies.” (Absolute Binding Free Energy Calculations: On the Accuracy of Computational Scoring of Protein-ligand Interactions, Singh and Warshel, Proteins, 2010). To validate the MM-PBSA results, the authors should employ an enhanced sampling method such as Hamiltonian replica exchange molecular dynamics, metadynamics, or umbrella sampling.

Author's reply:

For the free energy issue raised by the reviewer, again due to our computational limitation, we are not able to do this for all of the mutations. We have chosen two cases; one is the wild type structure i.e. where the codon is AUG and another is AUA which showed large penalty in MM-PBSA calculation for the one mutant case. We carried out well-tempered Metadynamics¹ to accelerate the conformational sampling along the codon-anticodon distance and to obtain the free energy landscape corresponding to the conformations sampled at different distances using the truncated structures. We used the Plumed code (ver. 2.0.2)² to incorporate the Metadynamics functionalities in Amber. We used a “DISTANCE” parameter as the collective variable. The width of the Gaussian (σ) was set to 0.02

nm for the CV with the initial height of Gaussians $W = 0.2$ KJ/mol added every 2 ps. The free energy surface for the two systems is shown in figure 4.

Figure 4: 1D Free energy Surface (FES) with distance between the center of mass (COM) distance of codon-anticodon for the two different codon systems (black for AUG and red when the codon is AUA). The snapshots showing the codon-anticodon interaction of the corresponding minimum is shown inset.

For the system having AUG as the codon, the most stable minimum of the FES corresponds to the codon-anticodon distance of ~ 0.8 nm, followed by another minimum at ~ 0.11 nm. The other system which has AUA as codon instead of AUG, the most probable minimum comes at the distance of ~ 1.6 nm, and the next stable minimum is at 1.25 nm. A small minimum for AUG is also observed at ~ 0.6 nm. This indicates that the most stable conformation for AUG codon comes almost at half distance than the same for AUA. Thus our metadynamics result shows that system having AUG codon has the most stable conformation when the codon-anticodon distance is smaller. Whereas when the codon is mutated to AUA, the stability arises when the codon-anticodon distance is higher compared to that in the case of AUG. This means the codon-anticodon interaction is not favourable in case of AUA compared to AUG as start codon. This strongly supports our MM-PBSA calculation which shows that for AUA, the codon-anticodon interaction is not stable and higher energetic penalty for AUA:CAU was found. Though, it is well known that MM-PBSA is not accurately produce binding free energy, we are able to validate our MM-PBSA calculations by the metadynamics simulation

References:

1. Barducci, A.; Bussi, G.; Parrinello, M. *Phys. Rev. Lett.* **2008**, 100, 1–4.
2. Tribello, G. A.; Bonomi, M.; Branduardi, D.; Camilloni, C.; Bussi, G. *Comput. Phys. Comm.* **2014**, 185, 604–613.

3. Upon close inspection of the manuscript, I am concerned about how the authors obtained their average structures. They state:

“So, we have extracted 40 frames of the last 40ns of the simulation trajectory for each run for single system and the snapshot of the averaged structures are represented in the figure S2.”

and

“We have extracted and averaged 40 frames of the last 40ns of the simulation trajectory for each run for a single system to obtain an average MD structure. It is now mentioned on Pg 17 in the manuscript.”

Simple averaging of structure coordinates can yield sterically unfeasible models. One proper way to do this would be to make histograms (or energy landscapes) and take a typical structure from the dominant basin, rather than averaging structure coordinates.

Authors reply: We completely agree with the reviewer’s concern about the average MD structure. But we have tested for AUG and AUA by superimposing the most populated structure with the average MD structure what shows the superimposition of the two structures in each case. And also for each case and for each run, the structures are not fluctuating much, so we believe our representative average MD structure are not unfeasible.

Reviewer #3 (Remarks to the Author):

The manuscript has improved a lot. In my opinion, all of the issues, except for the convergence issue, were sufficiently addressed by the authors.

In their reply, the authors claim that convergence is reached within 60 ns based on the rmsd plot (Fig. 5a). However, the rmsd of single trajectories does not suffice to test for convergence. For example, if the authors would have stopped their simulations after 8 ns, the rmsds would also look rather flat, because the increase in rmsd for the AUA simulation happens around 10 ns. Therefore, with the same reasoning, the authors could have claimed based on the 8 ns simulations that they were converged. This would obviously not have been true given the increase in rmsd that is seen at 10 ns with longer simulations.

It would be more convincing if they see similar levels of rmsd reached in the repetitions. I do not understand why replicates are not shown.

Authors reply: We are thankful to the referee for pointing this out. We have modified Figure 5a with all the replicate runs (4 for each) with error bars.

Figure 5a is replaced by the following figure:

Another argument for convergence presented by the authors in the reply is that the average structures of two runs of the GUG simulations are similar. Showing only these two examples is not convincing, because it is not clear why these were selected. Is this similarity observed for all repetitions of each codon?

Authors reply: The similarity was observed for all runs.

It is not yet clear to me how many simulations were run for each codons, in the methods part, the authors write that simulations were repeated four to six times, but in all the time traces (Figs. 5, S5, and S6) only one trace per codon is shown. How were they selected?

Authors reply: We are reporting four simulations in all cases and have modified the above-mentioned figures.

Figure 5a is be replaced by:

Figure S5 is replaced by:

Figure S6 is replaced by:

To improve transparency, the authors should provide a list of all simulations (Supplement) and also show the time traces for all simulations.

Authors reply: We have provided a list of all simulations as 2 tables in supplementary data (Supplementary tables 1 and 2).

Briefly, 4 set of independent simulation runs were carried out for each system and the runs were of length 60ns. For a few systems 6 set of runs were made however; we have considered only 4 runs in all calculations to make it uniform.

REVIEWERS' COMMENTS:

Reviewer #2 (Remarks to the Author):

1. The authors were asked to perform a simulation of the full complex for validation. They completed this and it validated their results.
2. The authors state that "Though, it is well known that MM-PBSA is not accurately produce binding free energy, we are able to validate our MM-PBSA calculations by the metadynamics simulation." The authors have validated two of their calculations using metadynamics.
3. The authors confirm that their average structures are similar to making histograms (or energy landscapes) and take a typical structure from the dominant basin, rather than averaging structure coordinates. The typical structures, rather than the average structures, should be used in the figures.

Reviewer #3 (Remarks to the Author):

The authors have addressed all my points raised during the review process and, in my view, the manuscript can be accepted.